# GT-MEAN LOSS: A SIMPLE YET EFFECTIVE SOLUTION FOR BRIGHTNESS MISMATCH IN LOW-LIGHT IMAGE ENHANCEMENT

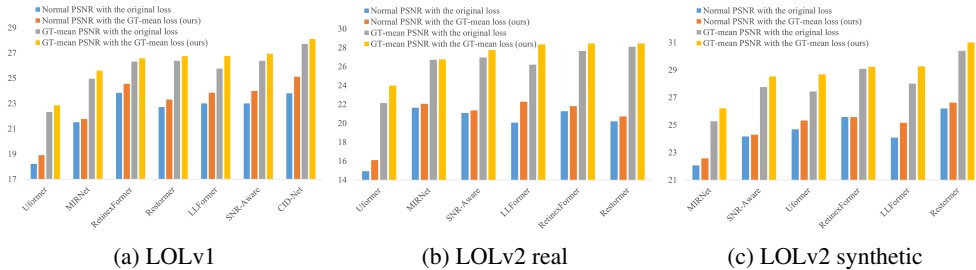

| (a) LOLv1 | (b) LOLv2 real | (c) LOLv2 synthetic |
|---|---|---|

Figure 1: Performance across various supervised LLIE models trained by their original loss functions and our GT-mean loss functions. The performance is consistently improved when the GT-mean loss functions are adopted. Notably, this improvement is easily attainable, as the use of GT-mean loss functions is flexible and brings minimal additional computational costs during training.

## ABSTRACT

Low-light image enhancement (LLIE) aims to improve the visual quality of images captured under poor lighting conditions. In supervised LLIE tasks, there exists a significant yet often overlooked inconsistency between the overall brightness of an enhanced image and its ground truth counterpart, referred to as *brightness mismatch* in this study. Brightness mismatch negatively impact supervised LLIE models by misleading model training. However, this issue is largely neglected in current research. In this context, we propose the *GT-mean loss*, a simple yet effective loss function directly modeling the mean values of images from a probabilistic perspective. The GT-mean loss is flexible, as it extends existing supervised LLIE loss functions into the GT-mean form with minimal additional computational costs. Extensive experiments demonstrate that the incorporation of the GT-mean loss results in consistent performance improvements across various methods and datasets.

## 1 INTRODUCTION

Low-light image enhancement (LLIE) is a crucial task in computer vision, aiming to improve the overall quality of images captured under poor lighting conditions (Li et al., 2022; Liu et al., 2021a). The primary objective of training a supervised LLIE model, denoted as $f(\cdot)$, is to map a low-light image $x$ to an enhanced image $f(x)$, subjecting to the constraint that $f(x)$ should resemble the ground truth (GT) image $y$ as much as possible. Under this paradigm, a well-trained LLIE model is expected to improve brightness while suppressing other degeneration factors commonly existed in low-light images, such as noise (Lu & Jung, 2022; Wei et al., 2020; Moseley et al., 2021), color distortion (Yan et al., 2024; Zhang et al., 2022), and others (Zhou et al., 2022; 2021).

In supervised LLIE tasks, the inconsistency between $f(x)$ and $y$ in terms of the overall brightness widely exists. In this paper, we refer to this widespread yet overlooked phenomenon as *brightness mismatch*. It can be simply represented as the inequality between the average brightness of $f(x)$ and $y$, i.e., $\mathbb{E}[f(x)] \neq \mathbb{E}[y]$. We find that this phenomenon can lead to biases in computing loss values

and evaluating visual quality, therefore negatively impacting the training phase and the evaluation phase of LLIE research.

**Impact on Evaluation.** Traditional metrics like PSNR can be biased by brightness mismatch. In Figure 2, we provide a typical example that the two modified images with vastly different visual quality still receive similar PSNR values. The primary reason for this inaccurate evaluation is that brightness mismatch dominates the PSNR values. This example indicates that traditional metrics, especially those directly based on pixel values, can be less comprehensive for evaluating visual quality at the presence of brightness mismatch.

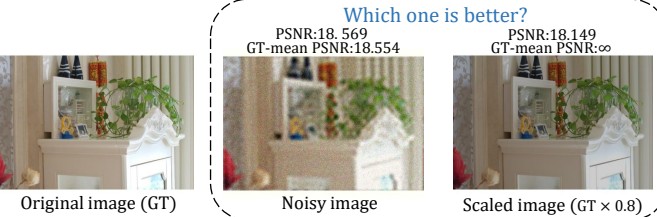

Figure 2: A comparison between the noisy image (obtained via by removing some Fourier high-frequency components and adding Gaussian noise) and the scaled image (the original image's brightness scaled by a factor of 0.8). Taking a zoomed-in view, it is obvious that the scaled image exhibits significantly better quality. However, the PSNR values of the two images are comparable, posing challenges to this most commonly used metric for evaluating LLIE's performance. In the contrary, their GT-mean PSNR values present huge difference, aligning with the true condition. For the scaled image, GT-mean PSNR is computed as $PSNR(\frac{\mathbb{E}[GT]}{\mathbb{E}[GT \times 0.8]}GT \times 0.8, GT)$, resulting in an infinite PSNR value (In practice, PSNR usually has a pre-defined upper bound. In this figure, we strictly follow the mathematical definition of PSNR for demonstration purposes).

**Impact on Training.** Brightness mismatch poses a risk of misleading model training. Table 1 illustrates that the low-quality noisy image is more likely to receive smaller loss values compared to the high-quality scaled image when tradition loss functions are employed. This example demonstrates that brightness mismatch can create an inaccurate association between loss value and visual quality during training. Grounded in the loss minimization paradigm, LLIE models are optimized to produce enhanced images with lower loss values, even if their visual quality is not guaranteed. This incorrect association can negatively impact model training, leading to less satisfying results.

Table 1: Loss values from the original losses and their GT-mean versions from the modified images from Figure 2. The experiment reveals an unintended behavior in the original loss functions, where most of them assign lower loss values to the low-quality image while assigning higher loss values to the high-quality image. The GT-mean loss functions, in contrast, successfully address this issue by correctly assigning lower loss values to the high-quality image and higher loss values to the low-quality image. The lower value between the noisy image and the scaled image is underlined.✗ indicating cases where the loss incorrectly identifies the noisy image as having better quality, and ✓ indicating cases where the model correctly identifies the scaled image as the better one.

| Loss type | Loss | Loss value | |
|---|---|---|---|
| | | Noisy image | Scaled image (×0.8) |
| Tradition loss | $L_1$ loss | 0.0912 ✗ | 0.1175 |
| | $L_2$ loss | 0.0138 ✗ | 0.0153 |
| | Perceptual loss | 5.6388 | 0.1119 ✓ |
| | Smooth $L_1$ loss | 0.0069 ✗ | 0.0076 |
| GT-mean loss | GT-mean $L_1$ loss | 0.0915 | 0.0193 ✓ |
| | GT-mean $L_2$ loss | 0.0138 | 0.0025 ✓ |
| | GT-mean Perceptual loss | 5.6466 | 0.0184 ✓ |
| | GT-mean Smooth $L_1$ loss | 0.0069 | 0.0012 ✓ |

As for model evaluation, (Wang et al., 2022a; Zhou et al., 2023; Jinhui et al., 2023; Yan et al., 2024) introduced GT-mean metrics to avoid the negative effects brought by brightness mismatch. The new evaluation metrics extend the original ones by aligning the average brightness of $\mathbb{E}[y]$ and $\mathbb{E}[f(x)]$ in advance. Specifically, the enhanced image is firstly rescaled as $\frac{\mathbb{E}[y]}{\mathbb{E}[f(x)]}f(x)$ to ensure that the evaluation is based on exactly the same average brightness. For example, the GT-mean PSNR metric can be obtained through $PSNR(\frac{\mathbb{E}[y]}{\mathbb{E}[f(x)]}f(x), y)$. From Figure 2, we can see that the GT-mean PSNR of the scaled image approaches infinity, showing the ideal fidelity between the scaled

image and GT after excluding brightness mismatch. Therefore, GT-mean metrics have the potential of cooperating with the original metrics for a comprehensive performance evaluation.

The issue of model training under brightness mismatch has largely been ignored in existing supervised LLIE research, despite some indirect solutions that do not primarily address this problem. For example, (Chen et al., 2018; Yang et al., 2021; Wu et al., 2022; Ma et al., 2023) designed multiple sub-networks to decouple brightness from other factors and optimized them separately. Nevertheless, the divide-and-conquer roadmap inevitably complicates the model design, as well as introducing significant computational overhead.

Inspired by the GT-mean metrics, we propose a simple yet effective loss function, called GT-mean loss, through explicitly modeling brightness mismatch in its construction. The loss dynamically balances its focus during training. For example, when $f(x)$ and $y$ are close, the loss function becomes unaware of brightness mismatch, and drives the model to focus more on optimizing various imaging factors except overall brightness. Therefore, the loss is able to eliminate the negative impact caused by brightness mismatch during training, facilitating LLIE models to comprehensively improve visual quality in a more effective way. The use of this loss function is straightforward, as it directly extends any existed loss function that requires the enhanced image $f(x)$ and its GT counterpart as inputs. The GT-mean loss is highlighted in the following aspects:

- **Simplicity**: The construction of the GT-mean loss is both theoretically and practically straightforward. Its underlying mechanism is easy to understand, and its implementation is uncomplicated.
- **Flexibility**: The GT-mean loss is highly flexible. For instance, the $L_1$ loss can be directly extended into the $L_1$ GT-mean version. This character makes adopting the GT-mean loss a universal choice for supervised LLIE models to upgrade their loss functions.
- **Low Cost**: Using the GT-mean loss introduces minimal overhead during training (approximately doubling the original loss computation). It is negligible compared to the overall model optimization process.
- **Effectiveness**: Extensive experiments have demonstrated that the GT-mean loss consistently improves model performance across a wide range of supervised LLIE methods (as shown in Figure 1).

## 2 BACKGROUND

### 2.1 SUPERVISED LLIE FRAMEWORK

In a supervised low-light image enhancement (LLIE) framework, the objective is to learn a mapping function $f$ that transforms a low-light image $x$ into an enhanced image $f(x)$ that closely approximates the ground truth $y$. This is typically achieved by minimizing a loss function that penalizes the differences between $f(x)$ and $y$, driving the model to produce outputs that have good visual quality as in ground truth images. Various loss functions can be applied for this task. Commonly used loss functions, such as the $L_1$ loss, are primarily adopted to minimize pixel-wise differences between the input image and the ground truth, ensuring the model generates accurate reconstructions.

### 2.2 LOSS FUNCTIONS FOR LLIE

Loss function is essential in LLIE tasks, as it directs model training. We categorize the loss functions in LLIE into two groups based on their purposes.

**Fidelity Losses.** These losses are designed to ensure that the enhanced image $f(x)$ closely resembles the ground truth $y$. They operate across various image representations, including pixel space, color space, frequency domain, and semantic space. For instance, the $L_1$-like loss functions directly ensure the pixel-level fidelity (Li et al., 2022; Liu et al., 2021a). Loss functions focused on color representation often use color histogram-based metrics (Yan et al., 2024), while others preserve fidelity at the frequency domain (Wang et al., 2023a; Huang et al., 2022). Additionally, some loss functions aim to maintain fidelity at higher representation levels, such as the perceptual loss (Johnson et al., 2016). Recently, some novel loss functions have emerged that subtly utilize fine-grained semantic information (Liang et al., 2023; Wu et al., 2023). Among these, pixel-level loss functions

are indispensable for image reconstruction. However, their effectiveness may be compromised by brightness mismatch.

**Prior-Based Losses.** These losses integrate domain-specific prior knowledge into LLIE models, aiming to maximize the use of available information. Typical examples are the Retinex-based methods(Zhang et al., 2019; Wei et al., 2018; Chen et al., 2018; Yang et al., 2021; Wu et al., 2022; Ma et al., 2023; Fu et al., 2023a), which are founded on the Retinex theory that an image is composed of illumination map and reflectance map. Specific loss functions are employed to penalize the locally smooth properties of illumination map and the lightness-insensitive properties of reflectance map. These loss functions are closely linked to specific model architectures and their underlying assumptions, limiting their generalizability. Furthermore, for unsupervised LLIE models that lack GT images for training, prior-based loss functions are essential for guiding model optimization. However, these loss functions can be less robust. For example, ZeroDCE (Li et al., 2021) builds an exposure control loss function with a hard threshold, which may result in over-exposure.

To pursue comprehensive visual quality enhancement, LLIE models tend to incorporate multiple loss functions. While this strategy can lead to improved results, it also increases the burden of model design and hyperparameter tuning. More importantly, as the existing loss functions overlook the brightness mismatch factor, the fundamental challenges brought by this factor remain unaddressed.

Our GT-mean loss is designed to directly address the issues caused by brightness mismatch. We argue that the primary goal of LLIE is to improve visibility while simultaneously suppressing other degenerated factors. Therefore, loss functions specifically designed for supervised LLIE should be sensitive to brightness mismatch and as concise as possible. To this end, we introduce brightness mismatch into the construction of GT-mean loss, and design a mechanism that dynamically balances the importance of optimizing for brightness and other image quality factors during training.

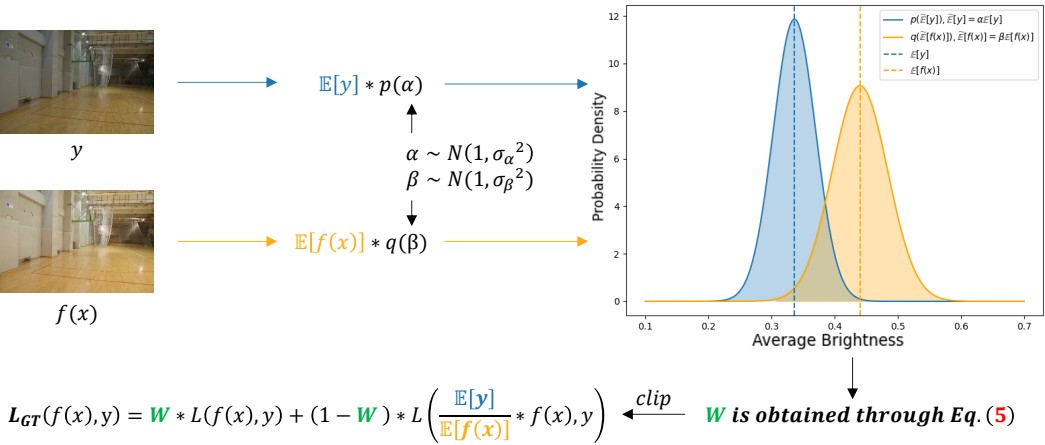

Figure 3: Illustration of the GT-mean loss construction. The average brightness values, $\mathbb{E}[y]$ and $\mathbb{E}[f(x)]$, are modeled as random variables $\widetilde{\mathbb{E}}[y] = \alpha\mathbb{E}[y]$ and $\widetilde{\mathbb{E}}[f(x)] = \beta\mathbb{E}[f(x)]$, where $\alpha \sim \mathcal{N}(1, \sigma_\alpha^2)$ and $\beta \sim \mathcal{N}(1, \sigma_\beta^2)$. The right side of the figure exemplifies the distributions $p(\widetilde{\mathbb{E}}[y])$ and $q(\widetilde{\mathbb{E}}[f(x)])]$ for both images. The GT-mean loss $L_{GT}$ combines the original loss $L(f(x), y)$ with a brightness-adjusted loss $L\left(\frac{\mathbb{E}[y]}{\mathbb{E}[f(x)]}f(x), y\right)$, weighted by $W$.

## 3 METHOD

In Section 3.1, we present the formulation of the GT-mean loss. In Section 3.2, we detail the crucial component of this loss function. Section 3.3 summarizes the features of the proposed loss.

### 3.1 GT-MEAN LOSS

In general, the GT-mean loss $L_{GT}(f(x), y)$ can be regarded as an extension of the existing loss $L(f(x), y)$ used for LLIE. To deal with the issues arising from brightness mismatch, the key to constructing $L_{GT}(f(x), y)$ is matching the average brightness of $f(x)$ and $y$. Furthermore, $L_{GT}(f(x), y)$ is designed to retain the form and effectiveness of the original loss $L(f(x), y)$. As illustrated in Figure 3, $L_{GT}(f(x), y)$ is formulated as follows:

$$L_{GT}(f(x), y) = W \cdot L(f(x), y) + (1 - W) \cdot L\left(\frac{\mathbb{E}[y]}{\mathbb{E}[f(x)]} f(x), y\right), \quad (1)$$

where $\frac{\mathbb{E}[y]}{\mathbb{E}[f(x)]}$ is a scaling factor for aligning the average brightness of $f(x)$ and $y$. The weight $W \in [0, 1]$ balances the two terms in $L_{GT}$. It is noted that the choice of $L(\cdot)$ is is arbitrary, provided that it accepts $f(x)$ and $y$ as inputs.

The primary strength of the GT-mean loss lies in its ability to dynamically balance the model's focus during training. In early stages, when the difference between $p(\widetilde{\mathbb{E}}[y])$ and $q(\widetilde{\mathbb{E}}[f(x)])$ is significant, $W$ is expected to approach 1 to make the first term $L(f(x), y)$ dominate the overall loss function. This behavior ensures that the GT-mean loss resembles the original loss $L(f(x), y)$, prioritizing improvements in overall image quality. As the training progresses and $p(\widetilde{\mathbb{E}}[y])$ and $q(\widetilde{\mathbb{E}}[f(x)])$ becomes closer, $W$ decreases to a smaller value. This trend shifts the emphasis of the overall loss function toward the second term $L\left(\frac{\mathbb{E}[y]}{\mathbb{E}[f(x)]} f(x), y\right)$, ensuring $f(x)$ and $y$ are compared under the condition of average brightness alignment. In this stage, GT mean loss mainly compares the image differences at the same mean brightness to avoid the negative effect of brightness mismatch, thus maintaining effective model training. From this mechanism, it is clear that $W$ plays a crucial role, which will be discussed in detail in the following subsection.

### 3.2 WEIGHT DESIGN

#### 3.2.1 PROBABILISTIC MODELING ON AVERAGE BRIGHTNESS

Instead of modeling the average brightness $\mathbb{E}[\cdot]$ as a fixed value, we represent it as a random variable, motivated by two key considerations. First, probabilistic modeling aligns well with the characteristics of human brightness perception. According to the Contrast Sensitivity Function (Robson, 1966; Bühren, 2018), human vision is highly sensitive to local contrasts, such as edges, textures, and intensity changes, but much less sensitive to a minor shift of $\mathbb{E}[\cdot]$. As long as local contrast remains intact, such minor shifts are unlikely to affect the human perception on visual quality, especially when $\mathbb{E}[\cdot]$ is relatively high. Second, probabilistic modeling enhances the control of our loss function during training. It allows us to estimate $W$ using common metrics like the Kullback-Leibler divergence or the Wasserstein distance. This facilitates a smooth weighting between the two terms in Eq. 1 without causing abrupt changes. Consequently, the loss exhibits good continuity as parameters change and has flat regions around its minima.

Based on these considerations, we regard $\mathbb{E}[\cdot]$ as an observation from a random variable $\widetilde{\mathbb{E}}[\cdot]$ obeying the Gaussian distribution. Specifically, $\widetilde{\mathbb{E}}[y]$ can be represented as:

$$\widetilde{\mathbb{E}}[y] = \alpha \mathbb{E}[y], \quad \alpha \sim \mathcal{N}(1, \sigma_\alpha^2), \quad (2)$$

where $\alpha$ determines the probability distribution type of $\widetilde{\mathbb{E}}[y]$, and $\sigma_\alpha^2$ defines its spread. Therefore, we have $p(\widetilde{\mathbb{E}}[y]) = \mathcal{N}(\mu_y, \sigma_y^2)$, with $\mu_y = \mathbb{E}[y]$ and $\sigma_y = \sigma_\alpha \mathbb{E}[y]$.

Similarly, $\mathbb{E}[f(x)]$ can be also seen as an observation from $\widetilde{\mathbb{E}}[f(x)]$, represented as:

$$\widetilde{\mathbb{E}}[f(x)] = \beta \mathbb{E}[f(x)], \quad \beta \sim \mathcal{N}(1, \sigma_\beta^2), \quad (3)$$

where $\beta$ determines the probability distribution type of $\widetilde{\mathbb{E}}[f(x)]$, and $\sigma_\beta^2$ defines its spread. Similarly, we have $q(\widetilde{\mathbb{E}}[f(x)]) = \mathcal{N}(\mu_{fx}, \sigma_{fx}^2)$, with $\mu_{fx} = \mathbb{E}[f(x)]$ and $\sigma_{fx} = \sigma_\beta \mathbb{E}[f(x)]$.

### 3.2.2 Estimation of $W$

We measure the difference between $p(\widetilde{\mathbb{E}}[y])$ and $q(\widetilde{\mathbb{E}}[f(x)])$ to estimate $W$ based on the Kullback-Leibler (KL) divergence. Instead of calculating $D_{KL}(p(\widetilde{\mathbb{E}}[y])||q(\widetilde{\mathbb{E}}[f(x)]))$ directly, we build an intermediate Gaussian distribution $\mathcal{N}(\mu_m, \sigma_m^2)$, and compute its distance to $\mathcal{N}(\mu_y, \sigma_y^2)$ and $\mathcal{N}(\mu_{fx}, \sigma_{fx}^2)$, respectively. Based on this, the weight $W$ can be estimated as:

$$W = \frac{1}{2} D_{KL}(\mathcal{N}(\mu_y, \sigma_y^2)||\mathcal{N}(\mu_m, \sigma_m^2)) + \frac{1}{2} D_{KL}(\mathcal{N}(\mu_{fx}, \sigma_{fx}^2)||\mathcal{N}(\mu_m, \sigma_m^2)), \quad (4)$$

where $\mu_m = \frac{\mu_y + \mu_{fx}}{2}, \sigma_m^2 = \frac{\sigma_y^2 + \sigma_{fx}^2}{2}$. Given the Gaussian distribution, Eq. 4 has an analytic solution (proof provided in Appendix A):

$$W = \frac{1}{2} \left[ \log \frac{\sigma_m}{\sigma_y} + \frac{\sigma_y^2 + (\mu_y - \mu_m)^2}{2\sigma_m^2} - \frac{1}{2} \right] + \frac{1}{2} \left[ \log \frac{\sigma_m}{\sigma_{fx}} + \frac{\sigma_{fx}^2 + (\mu_{fx} - \mu_m)^2}{2\sigma_m^2} - \frac{1}{2} \right]. \quad (5)$$

Finally, a clipping operation is applied to $W$ to restrict its value within $[0, 1]$.

In supervised LLIE, the enhanced image $f(x)$ should closely resemble the ground truth image $y$. Consequently, we assume that the shape of $p(\widetilde{\mathbb{E}}[y])$ and $q(\widetilde{\mathbb{E}}[f(x)])$ are similar. Based on this assumption, we equate $\sigma_\alpha^2$ and $\sigma_\beta^2$, setting them both to $\sigma^2$. In our experiments, we empirically set $\sigma$ as 0.1 for all the comparisons. In Section 4.3, we delve deeper into the influence of $\sigma$ on the performance of LLIE.

### 3.3 Discussion of the GT-mean loss

In application, flexibility is the primary advantage of the GT-mean loss, as it is independent of the model architecture and only requires the enhanced image $f(x)$ and the ground truth $y$ as inputs. Therefore, the GT-mean loss can be used in any supervised LLIE method by simply extending its supervised loss function into the GT-mean version.

The GT-mean loss is also highly efficient. Compared to the original loss, it doubles the computation, and $W$ is estimated with an analytical solution. Despite there is some extra computation overhead, the overall increase introduced by the GT-mean loss is negligible throughout the training process.

## 4 Experiment

### 4.1 Datasets and Settings

**Datasets.** We conducted experiments on both paired and unpaired datasets to evaluate our loss. For paired datasets, we used LOLv1 (Chen et al., 2018), LOLv2-real (Yang et al., 2021), and LOLv2-syn (Yang et al., 2021). Specifically, the LOLv1 dataset includes 485 training images and 15 testing images. The LOLv2-real dataset includes 689 training images and 100 testing images. The LOLv2-synthetic dataset includes 900 training images and 100 testing images. For unpaired datasets, we chose DICM (Lee et al., 2013), VV (Vonikakis et al., 2018), NPE (Wang et al., 2013), MEF (Ma et al., 2015), and LIME (Guo et al., 2017) as the test sets.

**Evaluation Metrics.** For the paired datasets, we used the normal evaluation metrics PSNR (Peak Signal-to-Noise Ratio) and SSIM (Structural Similarity Index) (Wang et al., 2004), along with GT-mean PSNR and GT-mean SSIM, to assess the effectiveness of the methods based on GT-mean loss.

For the unpaired datasets, we utilized three commonly used no-reference metrics, NIQE (Natural Image Quality Evaluator)(Mittal et al., 2013), BRISQUE (Blind/Referenceless Image Spatial Quality Evaluator)(Mittal et al., 2012), and PI (Perceptual Index)(Blau et al., 2018), to evaluate the performance.

**Baselines.** Seven supervised LLIE models were chosen as baselines. Their loss functions are shown in Table 2:

Table 2: Baselines and Their Loss Functions

| Method | Loss Function |
|---|---|
| Restormer (Zamir et al., 2022) | $L_1$ loss |
| RetinexFormer (Cai et al., 2023) | $L_1$ loss |
| LLFormer (Wang et al., 2023b) | Smooth $L_1$ loss (Girshick, 2015) |
| MIRNet (Zamir et al., 2020) | Charbonnier loss (Barron, 2019) |
| Uformer (Wang et al., 2022b) | Charbonnier loss |
| SNR-Aware (Xu et al., 2022) | Charbonnier loss, perceptual loss (Johnson et al., 2016) |
| CID-Net (Yan et al., 2024) | $L_1$ loss, edge loss (Seif & Androutsos, 2018), perceptual loss |

**Implementation Details.** To retrain the baselines equipped with GT-mean loss functions, we followed the their official settings, which can be found in Appendix C.

Table 3: Comparison on the Paired Datasets. + denotes the improvement of performance. The bold denotes the best among all the listed methods.

| Methods | Params/M | FLOPs/G | LOLv1 Normal PSNR | LOLv1 Normal SSIM | LOLv1 GT-mean PSNR | LOLv1 GT-mean SSIM | LOLv2-real Normal PSNR | LOLv2-real Normal SSIM | LOLv2-real GT-mean PSNR | LOLv2-real GT-mean SSIM | LOLv2-synthetic Normal PSNR | LOLv2-synthetic Normal SSIM | LOLv2-synthetic GT-mean PSNR | LOLv2-synthetic GT-mean SSIM |
|---|---|---|---|---|---|---|---|---|---|---|---|---|---|---|
| RetinexNet (Chen et al., 2018) | 0.84 | 587.47 | 16.774 | 0.419 | 18.935 | 0.427 | 16.097 | 0.401 | 18.323 | 0.447 | 17.137 | 0.762 | 19.099 | 0.774 |
| RUAS (Liu et al., 2021b) | 0.003 | 0.83 | 16.405 | 0.500 | 18.654 | 0.518 | 15.326 | 0.488 | 19.061 | 0.510 | 13.765 | 0.638 | 16.584 | 0.719 |
| EnlightenGAN (Jiang et al., 2021) | 114.35 | 61.01 | 17.480 | 0.651 | 20.003 | 0.691 | 18.640 | 0.675 | 21.434 | 0.675 | 16.572 | 0.774 | 19.493 | 0.825 |
| 3DLUT (Zeng et al., 2022) | 0.59 | 0.075 | 14.350 | 0.445 | 21.350 | 0.585 | 17.590 | 0.721 | 20.190 | 0.745 | 18.040 | 0.800 | 22.173 | 0.854 |
| ZeroDCE (Li et al., 2021) | 0.075 | 4.83 | 14.861 | 0.559 | 21.880 | 0.640 | 16.059 | 0.580 | 19.771 | 0.671 | 17.712 | 0.815 | 21.463 | 0.848 |
| Sparse (Yang et al., 2021) | 2.33 | 53.26 | - | - | - | - | 20.060 | 0.850 | 23.627 | 0.873 | 22.050 | 0.910 | 24.641 | 0.922 |
| PairLIE (Fu et al., 2023b) | 0.33 | 20.81 | 19.510 | 0.736 | 23.526 | 0.755 | 19.885 | 0.778 | 24.025 | 0.803 | - | - | - | - |
| Night Enhancement (Jin et al., 2022) | - | - | 21.521 | 0.768 | 24.231 | 0.781 | 20.850 | 0.724 | 25.447 | 0.796 | - | - | - | - |
| CUE (Zheng et al., 2023) | 0.25 | 157.32 | 21.680 | 0.774 | 24.700 | 0.794 | 22.562 | 0.803 | 27.626 | 0.832 | - | - | - | - |
| MTFE (Park et al., 2023) | - | - | 22.861 | 0.689 | 24.710 | 0.705 | - | - | - | - | - | - | - | - |
| IAT (Cui et al., 2022) | 0.09 | 5.28 | 23.382 | 0.808 | 25.275 | 0.815 | **23.499** | 0.824 | 27.248 | 0.836 | - | - | - | - |
| FourLLIE (Wang et al., 2023a) | - | - | - | - | - | - | 22.347 | 0.847 | 27.353 | 0.872 | 24.644 | 0.920 | 27.605 | 0.931 |
| Bread (Guo & Hu, 2022) | - | - | 20.620 | 0.834 | 25.299 | 0.846 | - | - | - | - | - | - | - | - |
| LEDNet (Zhou et al., 2022) | 7.07 | 35.92 | 20.627 | 0.823 | 25.470 | 0.846 | 19.938 | 0.827 | 27.814 | 0.870 | 23.709 | 0.914 | 27.367 | 0.928 |
| NeRCo (Yang et al., 2023) | 23.30 | - | 22.946 | 0.785 | 25.742 | 0.799 | - | - | - | - | - | - | - | - |
| FECNet (Huang et al., 2022) | 0.15 | - | 23.443 | 0.821 | 25.885 | 0.836 | - | - | - | - | - | - | - | - |
| MAXIM (Tu et al., 2022) | 14.1 | 216 | 23.435 | 0.864 | 27.555 | 0.877 | - | - | - | - | - | - | - | - |
| Uformer (Wang et al., 2022b) | 5.29 | 12 | 18.218 | 0.771 | 22.325 | 0.810 | 14.941 | 0.760 | 22.148 | 0.831 | 24.693 | 0.932 | 27.438 | 0.941 |
| Uformer with GT-mean loss (**ours**) | | | 18.915(+0.697) | 0.795 (+0.023) | 22.854(+0.529) | 0.830(+0.019) | 16.103(**+1.162**) | 0.792(+0.032) | 23.989(**+1.841**) | 0.858(+0.026) | 25.319(+0.626) | 0.940(+0.007) | 28.683(+1.245) | 0.948(+0.007) |
| MIRNet (Zamir et al., 2020) | 31.76 | 785 | 21.512 | 0.788 | 24.968 | 0.800 | 21.648 | 0.810 | 26.712 | 0.827 | 22.059 | 0.894 | 25.274 | 0.908 |
| MIRNet with GT-mean loss (**ours**) | | | 21.780(+0.268) | 0.804(+0.016) | 25.596(+0.628) | 0.818(+0.018) | 22.050(+0.402) | 0.830(+0.021) | 26.769(+0.057) | 0.846(+0.019) | 22.576(+0.517) | 0.906(+0.011) | 26.215(+0.941) | 0.918(+0.010) |
| RetinexFormer (Cai et al., 2023) | 1.53 | 15.57 | 23.830 | 0.832 | 26.312 | 0.844 | 21.272 | 0.841 | 27.650 | 0.877 | 25.281 | 0.928 | 28.827 | 0.939 |
| RetinexFormer with GT-mean loss (**ours**) | | | 24.561(+0.731) | 0.834(+0.003) | 26.586(+0.274) | 0.849(+0.005) | 21.810(+0.538) | **0.852**(+0.011) | 28.437(+0.787) | 0.879(+0.002) | 25.583(+0.299) | 0.933(+0.005) | 29.261(+0.434) | 0.944(+0.005) |
| Restormer (Zamir et al., 2022) | 26.13 | 144.25 | 22.718 | 0.830 | 26.375 | 0.848 | 20.235 | 0.848 | 28.159 | 0.880 | 26.288 | 0.944 | 30.570 | 0.955 |
| Restormer with GT-mean loss (**ours**) | | | 23.313(+0.595) | 0.837(+0.007) | 26.743(+0.368) | 0.855(+0.007) | 20.717(+0.482) | 0.845(+0.004) | **28.440**(+0.281) | **0.884**(+0.004) | **26.630**(+0.342) | **0.946**(+0.002) | **31.001**(+0.431) | **0.957**(+0.002) |
| LLFormer (Wang et al., 2023b) | 24.55 | 22.52 | 23.007 | 0.805 | 25.762 | 0.823 | 21.308 | 0.803 | 27.052 | 0.828 | 24.195 | 0.918 | 27.862 | 0.930 |
| LLFormer with GT-mean loss (**ours**) | | | 23.847(+0.840) | 0.830(**+0.025**) | 26.769(+1.007) | 0.846(+0.023) | 22.291(+0.983) | 0.844(**+0.041**) | 28.334(+1.282) | 0.870(**+0.420**) | 25.152(+0.957) | 0.932(**+0.014**) | 29.266(+1.404) | 0.945(**+0.015**) |
| SNR-Aware (Xu et al., 2022) | 4.01 | 26.35 | 23.005 | 0.824 | 26.373 | 0.843 | 21.103 | 0.839 | 26.971 | 0.866 | 24.173 | 0.924 | 27.756 | 0.937 |
| SNR-Aware with GT-mean loss (**ours**) | | | 23.992(+0.988) | 0.836(+0.012) | 26.942(+0.569) | 0.853(+0.009) | 21.350(+0.247) | 0.844(+0.005) | 27.740(+0.770) | 0.875(+0.010) | 24.301(+0.128) | 0.933(+0.009) | 28.525(+0.769) | 0.945(+0.008) |
| CID-Net (Yan et al., 2024) | 1.88 | 7.57 | 23.809 | 0.857 | 27.715 | 0.876 | - | - | - | - | - | - | - | - |
| CID-Net with GT-mean loss (**ours**) | | | **25.122**(+1.313) | **0.865**(+0.008) | **28.108**(+0.393) | **0.878**(+0.002) | - | - | - | - | - | - | - | - |

## 4.2 Quantitative Results

**Paired Datasets.** We present the performance of the GT-mean loss on the three paired datasets in Table 3.[1] We use two normal evaluation metrics, PSNR and SSIM, along with their GT-mean counterparts (GT-mean PSNR and GT-mean SSIM).

Among the seven baseline models, it is evident that the GT-mean loss consistently improves performance across all evaluation metrics, regardless of the type of the originally loss function. These results validate the effectiveness and flexibility of our GT-mean loss. The visual comparisons of paired datasets are provided in Appendix E. We note that using the GT-mean loss obviously does not alter the computational efficiency (in terms of FLOPs and Params) during the inference stage. Furthermore, considering the minimal additional computational overhead introduced during the training stage, the advantages provided by the GT-mean loss are easily attainable for supervised LLIE methods.

In addition to evaluating the selected baselines using GT-mean loss, Table 3 presents the performance of several previous LLIE methods. The objective is to demonstrate that GT-mean PSNR and GT-mean SSIM can serve as valuable complementary evaluation metrics for a comprehensively assessment of LLIE model performance. We can see that some methods, e.g., Bread and LEDNet, exhibit less satisfying PSNR and SSIM performance but achieve good performance when evaluated with GT-mean PSNR and GT-mean SSIM, showing the competitiveness of these methods. Since the GT-mean metric ensures that both images are compared at the same brightness level, it reduces the impact of brightness mismatch on the evaluation, placing greater emphasis on other visual quality factors, such as noise reduction and color distortions. In this context, we recommend reporting both GT-mean and normal metrics for a thorough performance evaluation, which will aid researchers in conducting in-depth analyses of how their models address the low-light image degradation factors.

---

[1] '-' in Table 3 indicates that these methods do not report the results or the officially released code does not work.

Moreover, as shown in Appendix B, the GT-mean metrics can also be used to determine the optimal stopping point during training, helping to prevent premature termination of the training process.

Table 4: Comparison on the Unpaired Datasets. $-(+)$ denotes the improvement(reduction) of performance. Note that all the models were trained on the LOLv2-synthetic dataset.

| Method | DICM | | | MEF | | | LIME | | | NPE | | | VV | | | AVG | | |
|---|---|---|---|---|---|---|---|---|---|---|---|---|---|---|---|---|---|---|
| | NIQE↓ | BRISQUE↓ | PI↓ | NIQE↓ | BRISQUE↓ | PI↓ | NIQE↓ | BRISQUE↓ | PI↓ | NIQE↓ | BRISQUE↓ | PI↓ | NIQE↓ | BRISQUE↓ | PI↓ | NIQE↓ | BRISQUE↓ | PI↓ |
| Restormer | 3.22 | 9.11 | 2.41 | 3.66 | 16.81 | 2.97 | 3.66 | 15.07 | 2.91 | 3.47 | 18.31 | 2.68 | 3.29 | 22.98 | 2.57 | 3.46 | 16.46 | 2.70 |
| Restormer with GT-mean loss (ours) | 3.18 | 8.79 | 2.36 | 3.63 | 17.11 | 2.85 | 3.63 | 17.11 | 2.91 | 3.45 | 18.49 | 2.68 | 3.3 | 22.05 | 2.56 | 3.44 (-) | 16.27(-) | 2.67(-) |
| MIRNET | 3.82 | 17.21 | 2.64 | 3.67 | 22.69 | 3.22 | 4.23 | 16.66 | 3.32 | 3.47 | 16.80 | 2.61 | 3.64 | 19.73 | 2.57 | 3.77 | 18.62 | 2.87 |
| MIRNET with GT-mean loss (ours) | 3.20 | 11.95 | 2.32 | 3.60 | 22.06 | 3.19 | 4.33 | 18.95 | 3.30 | 3.50 | 17.24 | 2.62 | 3.71 | 19.71 | 2.61 | 3.67(-) | 17.98(-) | 2.81(-) |
| Retinexformer | 3.23 | 9.99 | 2.37 | 3.86 | 15.08 | 3.04 | 3.88 | 13.59 | 2.98 | 3.38 | 16.16 | 2.62 | 2.73 | 14.51 | 3.27 | 3.42 | 13.87 | 2.86 |
| Retinexformer with GT-mean loss (ours) | 3.21 | 10.17 | 2.37 | 3.82 | 15.37 | 3.08 | 3.84 | 13.78 | 2.85 | 3.37 | 16.72 | 2.63 | 2.77 | 15.67 | 3.24 | 3.40(-) | 14.34(+) | 2.83(-) |
| SNR | 6.07 | 32.48 | 4.53 | 4.27 | 27.17 | 3.73 | 6.06 | 34.18 | 4.67 | 6.47 | 36.41 | 4.83 | 11.52 | 77.97 | 9.22 | 6.88 | 41.64 | 5.39 |
| SNR with GT-mean loss (ours) | 6.12 | 32.11 | 4.55 | 4.26 | 26.72 | 3.70 | 6.11 | 34.62 | 4.67 | 6.46 | 36.43 | 4.84 | 11.55 | 78.06 | 9.23 | 6.90(+) | 41.59(-) | 5.39 |
| Uformer | 3.08 | 8.45 | 2.38 | 3.72 | 13.63 | 2.87 | 3.66 | 11.45 | 2.81 | 3.40 | 15.96 | 2.67 | 2.70 | 16.02 | 3.20 | 3.31 | 13.10 | 2.79 |
| Uformer with GT-mean loss (ours) | 3.12 | 7.29 | 2.30 | 3.69 | 12.64 | 2.88 | 3.64 | 12.31 | 2.77 | 3.38 | 16.36 | 2.65 | 2.70 | 16.58 | 3.19 | 3.30(-) | 13.04(-) | 2.76(-) |
| LLformer | 3.26 | 15.04 | 2.45 | 3.75 | 21.16 | 2.93 | 4.01 | 17.08 | 2.94 | 3.32 | 15.02 | 2.62 | 3.16 | 12.32 | 2.43 | 3.50 | 16.13 | 2.68 |
| LLformer with GT-mean loss (ours) | 3.05 | 11.06 | 2.36 | 3.65 | 19.60 | 2.90 | 4.07 | 16.45 | 2.97 | 3.33 | 12.43 | 2.65 | 2.99 | 10.42 | 2.33 | 3.41(-) | 13.99(-) | 2.64(-) |

**Unpaired Datasets.** Table 4 presents the model performance across five unpaired dataset. Compared with the baseline performance, using GT-mean loss demonstrates superior or comparable results in most cases in terms of the three non-reference evaluation metrics. The findings on these unpaired datasets empirically highlight the generalization capability of GT-mean loss, as using this loss still yields performance improvements when tested on unseen images. For visual comparison, we randomly selected two images for each baseline, which can be found in Appendix E.

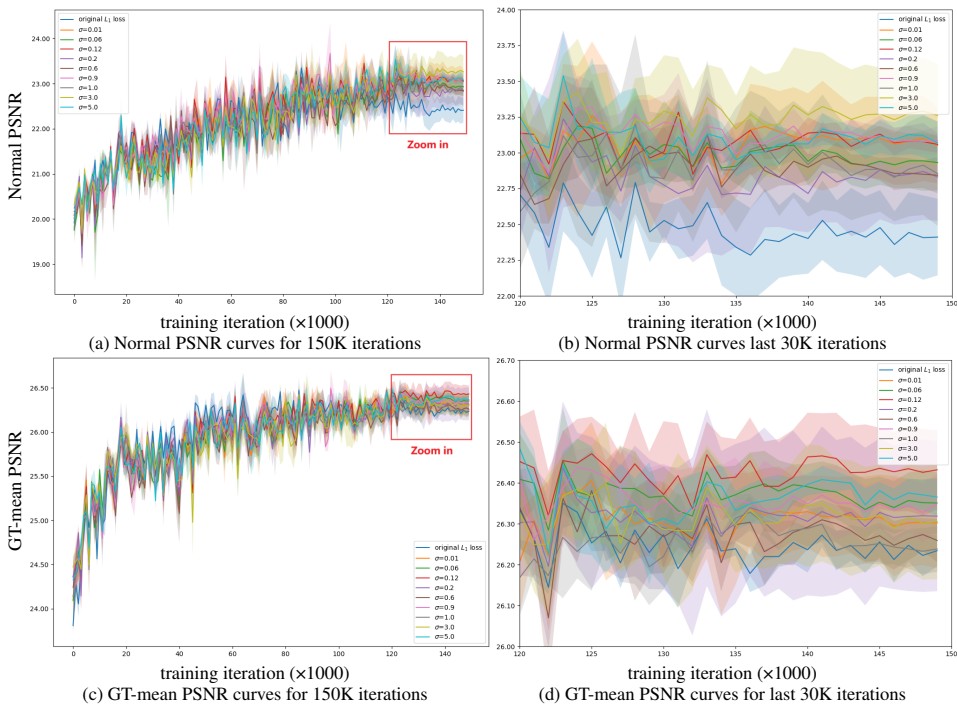

Figure 4: The effect of different $\sigma$ on model performance.

### 4.3 EFFECT OF THE PARAMETER $\sigma$

To investigate the influence of $\sigma$, we conducted experiments on LOLv1 using RetinexFormer (Cai et al., 2023) trained with GT-mean $L_1$ loss under different $\sigma$ values. We selected 10 different $\sigma$ values, running each setting three times for consistency. Notably, $\sigma = 0$ represents a special case where the GT-mean $L_1$ loss degrades to the original $L_1$ loss. For every 1,000 (1K) iterations in the 150K iterations, we calculated mean and variance of the normal PSNR and GT-mean PSNR values, shown in Figure 4 for demonstrating the trend during training. In the early stages (as can be seen in Figure 4 (a) and (c)), the curve tendencies under different $\sigma$ settings are similar. Considering the curve with $\sigma = 0$ closely resembles the original $L_1$ loss, we can empirically verify that the GT-mean loss at the early stage behaves like the original $L_1$ loss. In contrast, as shown in the zoomed-in

views of the last 30K iterations (Figure 4 (b) and (d)), we observe that all settings become stable, and the settings with non-zero $\sigma$ consistently perform better than $\sigma = 0$. This observation shows that the GT-mean loss diverges significantly from the original $L_1$ loss in the late training stage. The second term in Eq.1 allows the GT-mean loss to continuously improve model performance. In addition, the experiment shows that the choice of $\sigma$ value is open. As $\sigma$ measures the spread of the random variable $\widetilde{\mathbb{E}}[\cdot]$ deviating from the observed average image brightness $\mathbb{E}[\cdot]$ in our modeling, we recommend using a small value, such as $\sigma = 0.1$ for real world application.

## 4.4 FURTHER ANALYSIS ON THE GT-MEAN LOSS

In this experiment, we further investigate the difference between the original loss and the GT-mean loss. Specifically, we randomly selected a batch of low-light images (batchsize = 8) and their corresponding ground truth images. These images were enhanced using RetinexFormer to produce enhanced outputs $f(x)$. To simulate the varying brightness, we introduced a unified scaling factor $\eta$ ranging from 0 to 3, simulating the progression of the enhanced images $\eta \cdot f(x)$ from dark to bright. This experiment setting simulates how the loss value varies under different brightness levels, facilitating us to investigate the loss curve with respect to the brightness variation only.

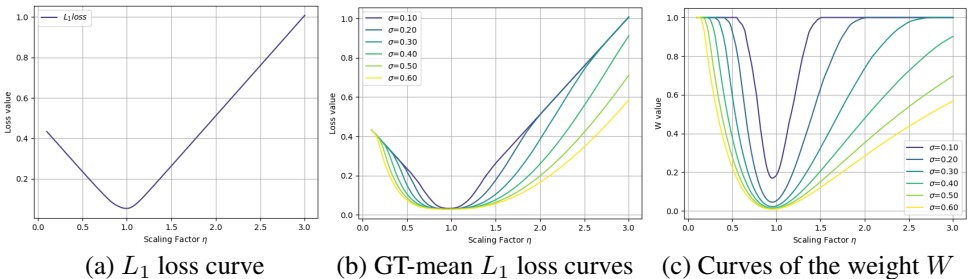

(a) $L_1$ loss curve     (b) GT-mean $L_1$ loss curves     (c) Curves of the weight $W$

Figure 5: Loss curves and weight curves for analyzing the effectiveness of the GT-mean loss.

Based on the above experimental design, we present the curve of the original $L_1$ loss (Figure 5(a)), and the curves of the GT-mean $L_1$ loss under different $\sigma$ values (Figure 5(b)). The difference between them is that the use of the GT-mean loss clearly produces basins around $\eta = 1$. In another word, the GT-mean loss produces small-gradient region with regard to brightness around $\eta = 1$, of which the range is controlled by $\sigma$. From an optimization perspective, since the gradients with respect to brightness become smaller, the optimization along the direction of brightness adjustment is in turn slowed down. Based on this characteristic, in real-world model training, the GT-mean $L_1$ loss enables LLIE models to focus on other important degeneration factors, when $p(\widetilde{\mathbb{E}}[y])$ and $q(\widetilde{\mathbb{E}}[f(x)])]$ become closer. In contrast, the original $L_1$ loss is less capable of decoupling the optimization with respect to brightness and other visual quality factors.

Additionally, Figure 5(c) presents the weight curves that correspond to Figure 5(b), demonstrating how the GT-mean $L_1$ loss behaves with regard to weight variation. As $\eta$ approaches 1, the weight $W$ rapidly decreases, indicating that the second term in Eq.1 begins to dominate the loss function, confirming the mechanism of our loss. Notably, as $\sigma$ increases, $W$ starts to drop at smaller values of $\eta$, meaning that the second term in Eq.1 takes over earlier in the optimization process. This behavior aligns with the design of $\sigma$, which controls the spread of $\widetilde{\mathbb{E}}[\cdot]$.

## 5 CONCLUSION

In this paper, we propose the GT-mean loss to advance research on supervised low-light image enhancement (LLIE) methods. The GT-mean loss enables the model training process to circumvent the misleading issue caused by brightness mismatch, thereby comprehensively addressing the various degeneration factors in low-light images. Due to its simple construction, the GT-mean loss can be easily adopted by existing supervised LLIE methods, imposing minimal additional computational overhead during training. Experiments across various supervised LLIE methods consistently demonstrate the effectiveness of the proposed loss. While the estimation of the weight $W$ remains

an open problem, we plan to explore various $W$-estimation strategies to potentially unlock even greater performance gains in LLIE models.

Additionally, we encourage the LLIE research community to adopt GT-mean metrics as a complement to traditional evaluation metrics. By incorporating traditional metrics alongside their GT-mean extensions, researchers can gain a comprehensive perspective on assessing the visual quality, thereby facilitating the development of more effective LLIE techniques.

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

## A   PROOFS

**Lemma 1.** *Let $p(x) = \mathcal{N}(\mu_1, \sigma_1)$ and $q(x) = \mathcal{N}(\mu_2, \sigma_2)$. According to the definition of KL divergence, we have:*

$$D_{KL}(p, q) = \mathbb{E}_p\left(\log\frac{p(x)}{q(x)}\right) = -\int p(x)\log q(x)\,dx + \int p(x)\log p(x)\,dx$$

$$= \log\frac{\sigma_2}{\sigma_1} + \frac{1}{2\sigma_2^2}\left(\sigma_1^2 + (\mu_1 - \mu_2)^2\right) - \frac{1}{2}$$

*Proof.* The KL divergence expression for two normal distributions can be written as:

$$D_{KL}(p, q) = \int p(x) \cdot \log\frac{\sigma_2}{\sigma_1}\exp\left(\frac{(x - \mu_2)^2}{2\sigma_2^2} - \frac{(x - \mu_1)^2}{2\sigma_1^2}\right)dx$$

$$= \mathbb{E}_p\left[\log\frac{\sigma_2}{\sigma_1} + \frac{1}{2}\left(\frac{(X - \mu_1)^2}{\sigma_1^2} - \frac{(X - \mu_2)^2}{\sigma_2^2}\right)\right]$$

$$= \log\frac{\sigma_2}{\sigma_1} + \frac{1}{2\sigma_2^2}\mathbb{E}_p\left[(X - \mu_2)^2 - (X - \mu_1)^2\right]$$

$$= \log\frac{\sigma_2}{\sigma_1} + \frac{1}{2\sigma_2^2}\mathbb{E}_p[(X - \mu_2)^2] - \frac{1}{2\sigma_1^2}\mathbb{E}_p[(X - \mu_1)^2]$$

$$= \log\frac{\sigma_2}{\sigma_1} + \frac{1}{2\sigma_2^2}\mathbb{E}_p[(X - \mu_2)^2] - \frac{1}{2} \tag{6}$$

We expand $(X - \mu_2)^2$ as follows:

$$(X - \mu_2)^2 = (X - \mu_1 + \mu_1 - \mu_2)^2 = (X - \mu_1)^2 + 2(X - \mu_1)(\mu_1 - \mu_2) + (\mu_1 - \mu_2)^2$$

Taking the expectation under $p$, we get:

$$\mathbb{E}_p\left[(X - \mu_2)^2\right] = \mathbb{E}_p\left[(X - \mu_1)^2\right] + 2(\mu_1 - \mu_2)\mathbb{E}_p\left[X - \mu_1\right] + (\mu_1 - \mu_2)^2$$

Since $\mathbb{E}_p\left[X - \mu_1\right] = 0$, this simplifies to:

$$\mathbb{E}_p\left[(X - \mu_2)^2\right] = \mathbb{E}_p\left[(X - \mu_1)^2\right] + (\mu_1 - \mu_2)^2 = \sigma_1^2 + (\mu_1 - \mu_2)^2$$

Now substituting this back into Eq. 6:

$$D_{KL}(p, q) = \log\frac{\sigma_2}{\sigma_1} + \frac{1}{2\sigma_2^2}\left(\sigma_1^2 + (\mu_1 - \mu_2)^2\right) - \frac{1}{2} \tag{7}$$

This concludes the proof.   $\square$

**Lemma 2.** *Let $p(x) = \mathcal{N}(\mu_1, \sigma_1)$ and $q(x) = \mathcal{N}(\mu_2, \sigma_2)$. According Eq.5, we have:*

$$W = \frac{1}{2}\left[\log\frac{\sigma_m}{\sigma_1} + \frac{\sigma_1^2 + (\mu_1 - \mu_m)^2}{2\sigma_m^2} - \frac{1}{2}\right] + \frac{1}{2}\left[\log\frac{\sigma_m}{\sigma_2} + \frac{\sigma_2^2 + (\mu_2 - \mu_m)^2}{2\sigma_m^2} - \frac{1}{2}\right]. \tag{8}$$

*where $\mu_m = \frac{\mu_1 + \mu_2}{2}, \sigma_m^2 = \frac{\sigma_1^2 + \sigma_2^2}{2}$.*

*We proof Eq.8 can be written as*

$$W = \frac{1}{4}\frac{(\mu_1 - \mu_2)^2}{\sigma_1^2 + \sigma_2^2} + \frac{1}{2}\log\left(\frac{\sigma_1^2 + \sigma_2^2}{2\sigma_1\sigma_2}\right), \tag{9}$$

*where Eq.9 is also the closed form of the Bhattacharyya distance after two one-dimensional Gaussian distributions (Kashyap, 2019).*

*Proof.* We can extend the first term in Eq.8 as:

$$\frac{1}{2}\left[\log\frac{\sigma_m}{\sigma_1} + \frac{\sigma_1^2 + (\mu_1 - \mu_m)^2}{2\sigma_m^2} - \frac{1}{2}\right] = \frac{1}{2}\log\frac{\sigma_m}{\sigma_1} + \frac{1}{2}\cdot\frac{\sigma_1^2 + (\mu_1 - \mu_m)^2}{2\sigma_m^2} - \frac{1}{4}. \tag{10}$$

Similarly, the second term can be extended as:

$$\frac{1}{2}\left[\log\frac{\sigma_m}{\sigma_2} + \frac{\sigma_2^2 + (\mu_2 - \mu_m)^2}{2\sigma_m^2} - \frac{1}{2}\right] = \frac{1}{2}\log\frac{\sigma_m}{\sigma_2} + \frac{1}{2}\cdot\frac{\sigma_2^2 + (\mu_2 - \mu_m)^2}{2\sigma_m^2} - \frac{1}{4}. \tag{11}$$

Combine this two term, we have:

$$\frac{1}{2}\log\frac{\sigma_m}{\sigma_1} + \frac{1}{2}\cdot\frac{\sigma_1^2 + (\mu_1 - \mu_m)^2}{2\sigma_m^2} - \frac{1}{4} + \frac{1}{2}\log\frac{\sigma_m}{\sigma_2} + \frac{1}{2}\cdot\frac{\sigma_2^2 + (\mu_2 - \mu_m)^2}{2\sigma_m^2} - \frac{1}{4}$$

$$= \frac{1}{2}\log\frac{\sigma_m}{\sigma_1} + \frac{1}{2}\log\frac{\sigma_m}{\sigma_2} + \frac{1}{2}\cdot\frac{\sigma_1^2 + (\mu_1 - \mu_m)^2}{2\sigma_m^2} + \frac{1}{2}\cdot\frac{\sigma_2^2 + (\mu_2 - \mu_m)^2}{2\sigma_m^2} - \frac{1}{2}$$

$$= \frac{1}{2}\log\frac{\sigma_m^2}{\sigma_1\sigma_2} + \frac{1}{2}\cdot\frac{\sigma_1^2 + \sigma_2^2 + (\mu_1 - \mu_m)^2 + (\mu_2 - \mu_m)^2}{2\sigma_m^2} - \frac{1}{2}. \tag{12}$$

Due to $\mu_m = \frac{\mu_1 + \mu_2}{2}, \sigma_m^2 = \frac{\sigma_1^2 + \sigma_2^2}{2}$, Eq.12 can be written as:

$$\frac{1}{2}\log\frac{\sigma_m^2}{\sigma_1\sigma_2} + \frac{1}{2}\cdot\frac{\sigma_1^2 + \sigma_2^2 + \frac{(\mu_1 - \mu_2)^2}{2}}{2\sigma_m^2} - \frac{1}{2}$$

$$= \frac{1}{2}\log\frac{\sigma_1^2 + \sigma_2^2}{2\sigma_1\sigma_2} + \frac{1}{2}\cdot\frac{\sigma_1^2 + \sigma_2^2 + \frac{(\mu_1 - \mu_2)^2}{2}}{\sigma_1^2 + \sigma_2^2} - \frac{1}{2}$$

$$= \frac{1}{4}\frac{(\mu_1 - \mu_2)^2}{\sigma_1^2 + \sigma_2^2} + \frac{1}{2}\log\frac{\sigma_1^2 + \sigma_2^2}{2\sigma_1\sigma_2} \tag{13}$$

This concludes the proof. $\square$

# B  GT-MEAN METRIC FOR TRAINING GUIDANCE

In low-light image enhancement (LLIE) training, determining the optimal number of iterations is challenging due to fluctuating performance and the risk of overfitting. Here, we demonstrate how the GT-mean metric can assist in identifying the optimal stopping point.

We saved RetinexFormer results every 1,000 (1K) iterations and evaluated them using normal metrics (PSNR and SSIM) and GT-mean metrics. Figure (a) shows normal metrics, where the PSNR curve flattens between 60k and 100k iterations, suggesting this as the optimal range. However, in Figure (b) (GT-mean metrics), the PSNR continues improving beyond 100k iterations, indicating further gains.

The GT-mean metrics provide consistent results across both PSNR and SSIM, unlike normal metrics, which show inconsistencies. This inconsistency in normal metrics could lead to suboptimal decisions regarding when to stop training. Thus, the GT-mean metric offers a clearer view of model improvement, helping select better training parameters and preventing premature termination due to concerns about overfitting.

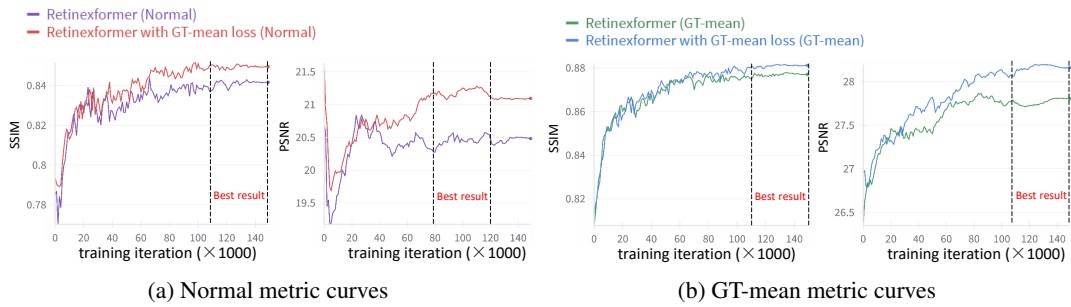

(a) Normal metric curves        (b) GT-mean metric curves

Figure 6: Metric curves during the training process. We evaluated the Normal metric and GT-mean metric every 1K iterations (out of a total of 150K iterations), with metrics including SSIM and PSNR.

## C EXPERIMENTAL DETAILS

In this section, we present the experimental setup for each method. Our aim is to ensure consistency with the official settings for each baseline model while introducing the GT-mean loss to demonstrate its effectiveness. To ensure fair comparisons, both the baseline models and the ones using GT-mean loss were trained under identical hardware and software environments, minimizing the effects of randomness.

**Uformer.** Both the baseline and the GT-mean loss variant were trained following the experimental setup for motion deblurring in (Wang et al., 2022b), selected Uformer-T as the backbone model. The Charbonnier loss used in the baseline was extended to GT-mean loss for the variant.

**MIRNet.** Both the baseline and the GT-mean loss variant were trained according to the settings used for the denoising task in (Zamir et al., 2020). In the GT-mean loss variant, the Charbonnier loss was replaced with the GT-mean loss.

**RetinexFormer.** For both the baseline and the GT-mean loss variant, we followed the training settings for LOL datasets in (Cai et al., 2023). The $L_1$ loss used in the baseline was extended to GT-mean loss in the variant.

**Restormer.** The baseline and the GT-mean loss variant were both trained following the motion deblurring settings described in (Zamir et al., 2022). The $L_1$ loss in the baseline was extended to GT-mean loss in the variant.

**LLFormer.** Both the baseline and the GT-mean loss variant were trained according to the settings for the LOLv1 dataset described in (Wang et al., 2023b). The Smooth $L_1$ loss used in the baseline was extended to GT-mean loss for the variant.

**SNR-Aware.** The baseline and the GT-mean loss variant were both trained using the settings for for LOL datasets outlined in (Xu et al., 2022). The Charbonnier loss and perceptual loss used in the baseline were extended to GT-mean loss in the variant.

**CID-Net.** Both the baseline and the GT-mean loss variant were trained using the LOLv1 settings described in (Yan et al., 2024). In the GT-mean loss variant, the Charbonnier loss, edge loss, and perceptual loss were extended to GT-mean loss.

In summary, for each method, the original loss functions were extended to GT-mean loss, and all models were trained using consistent settings to ensure a fair comparison.

# D MORE QUANTITATIVE RESULTS

We provide additional metrics to demonstrate the effectiveness of GT-mean loss. We have added Q-Align(Wu et al., 2024) for metric evaluation across all datasets, which includes two metrics: Image Quality Assessment (IQA) and Image Aesthetic Assessment (IAA), with a range of [0, 5], where higher values are better. We write it as IQA/IAA.

Additionally, for the paired datasets (as shown in Table.5), we supplement the normal Lpips(Zhang et al., 2018) and the GT-mean Lpips, where lower values are better. For the unpaired datasets (as shown in Table.6, we supplement MUSIQ(Ke et al., 2021), where higher values are better. Our approach achieves consistent improvement across all metrics.

Table 5: Lpips and Q-Align for on the Paired Datasets. For Lpips, ↓(↑) denotes the improvement(reduction) of performance. For Q-Align, ↑(↓) denotes the improvement(reduction) of performance.

| Method | LOLv1 | | | LOLv2-real | | | LOLv2-synthetic | | |
|---|---|---|---|---|---|---|---|---|---|
| | Normal Lpips↓ | GT-mean Lpips↓ | IQA/IAA↑ | Normal Lpips↓ | GT-mean Lpips↓ | IQA/IAA↑ | Normal Lpips↓ | GT-mean Lpips↓ | IQA/IAA↑ |
| RetinexFormer | 0.141 | 0.134 | 3.317/1.959 | 0.163 | 0.152 | 3.478/2.009 | 0.064 | 0.057 | 3.148/2.114 |
| RetinexFormer with GT-meanloss | 0.138↓ | 0.132↓ | 3.331/1.971 ↑ | 0.143↓ | 0.134↓ | 3.778/2.048 ↑ | 0.063↓ | 0.056↓ | 3.191/2.144 ↑ |
| MIRNet | 0.222 | 0.216 | 2.917/1.745 | 0.313 | 0.303 | 2.598/1.520 | 0.122 | 0.114 | 2.956/2.145 |
| MIRNet with GT-meanloss | 0.196↓ | 0.189↓ | 3.039/1.758 ↑ | 0.214↓ | 0.208↓ | 2.924/1.702 ↑ | 0.104↓ | 0.094↓ | 3.064/2.187↑ |
| LLFormer | 0.183 | 0.178 | 3.027/1.800 | 0.248 | 0.236 | 2.714/1.590 | 0.07 | 0.064 | 3.102/2.099 |
| LLFormer with GT-meanloss | 0.138↓ | 0.133↓ | 3.373/1.956↑ | 0.166↓ | 0.156↓ | 3.206/1.884↑ | 0.058↓ | 0.051↓ | 3.197/2.130↑ |
| Restormer | 0.128 | 0.122 | 3.567/2.032 | 0.162 | 0.147 | 3.478/1.987 | 0.045 | 0.039 | 3.350/2.187 |
| Restormer with GT-meanloss | 0.122↓ | 0.117↓ | 3.672/2.054↑ | 0.149↓ | 0.135↓ | 3.554/2.020↑ | 0.041↓ | 0.036↓ | 3.404/2.218↑ |
| Uformer | 0.212 | 0.195 | 3.087/1.946 | 0.228 | 0.199 | 2.882/1.827 | 0.06 | 0.055 | 3.176/2.137 |
| Uformer with GT-meanloss | 0.168↓ | 0.157↓ | 3.419/2.049 ↑ | 0.180↓ | 0.156↓ | 3.104/1.880 ↑ | 0.049↓ | 0.045↓ | 3.283/2.177↑ |
| SNR-Aware | 0.164 | 0.158 | 3.330/1.893 | 0.169 | 0.161 | 3.354/1.879 | 0.064 | 0.058 | 3.275/2.209 |
| SNR-Aware with GT-meanloss | 0.158↓ | 0.153↓ | 3.509/1.913↑ | 0.164↓ | 0.154↓ | 3.468/1.889 ↑ | 0.057↓ | 0.050↓ | 3.326↑/2.207 ↓ |
| CID-Net | 0.086 | 0.079 | 4.087/2.157 | | | | | | |
| CID-Net with GT-meanloss | 0.081↓ | 0.075↓ | 4.074↓/2.161↑ | | | | | | |

Table 6: Musiq and Q-Align for Five unpaired datasets. ↑(↓) denotes the improvement(reduction) of performance.

| Method | DICM | | MEF | | LIME | | NPE | | VV | | AVG | |
|---|---|---|---|---|---|---|---|---|---|---|---|---|
| | MUSIQ↑ | IQA/IAA↑ | MUSIQ↑ | IQA/IAA↑ | MUSIQ↑ | IQA/IAA↑ | MUSIQ↑ | IQA/IAA↑ | MUSIQ↑ | IQA/IAA↑ | MUSIQ↑ | IQA/IAA↑ |
| RetinexFormer | 57.398 | 3.800/2.740 | 56.17 | 3.111/2.323 | 57.262 | 3.111/2.323 | 60.507 | 3.673/2.699 | 37.513 | 3.471/2.154 | 53.770 | 3.438/2.458 |
| RetinexFormer with GT-meanloss | 57.247 | 3.805/2.773 | 56.633 | 3.273/2.423 | 57.374 | 3.273/2.423 | 60.682 | 3.706/2.719 | 37.654 | 3.517/2.166 | 53.918↑ | 3.490/2.498↑ |
| MIRNet | 52.467 | 3.111/2.337 | 47.399 | 2.860/2.088 | 54.837 | 2.860/2.088 | 58.641 | 3.285/2.374 | 54.566 | 2.955/2.162 | 53.582 | 2.991/2.203 |
| MIRNet with GT-meanloss | 53.188 | 3.295/2.375 | 47.611 | 2.747/2.058 | 55.776 | 2.747/2.058 | 58.718 | 3.366/2.428 | 54.891 | 3.120/2.215 | 54.037↑ | 3.069/2.225↑ |
| LLFormer | 56.642 | 3.379/2.526 | 53.335 | 2.836/2.102 | 55.671 | 2.836/2.102 | 59.824 | 3.445/2.551 | 60.885 | 3.067/1.955 | 57.271 | 3.079/2.225 |
| LLFormer with GT-meanloss | 57.038 | 3.521/2.571 | 53.842 | 2.946/2.152 | 55.83 | 2.946/2.152 | 60.044 | 3.580/2.605 | 60.858 | 3.137/1.997 | 57.522↑ | 3.178/2.268↑ |
| Restormer | 58.525 | 3.885/2.800 | 56.528 | 3.267/2.466 | 58.461 | 3.267/2.466 | 61.031 | 3.781/2.735 | 37.919 | 3.710/2.264 | 54.493 | 3.572/2.536 |
| Restormer with GT-meanloss | 58.604 | 3.913/2.821 | 56.522 | 3.380/2.521 | 58.124 | 3.380/2.521 | 60.971 | 3.820/2.769 | 38.29 | 3.712/2.255 | 54.502↑ | 3.607/2.559↑ |
| Uformer | 58.084 | 3.832/2.788 | 56.177 | 3.040/2.343 | 57.698 | 3.040/2.343 | 61.31 | 3.657/2.716 | 36.235 | 3.557/2.249 | 53.901 | 3.453 /2.500 |
| Uformer with GT-meanloss | 58.981 | 3.910/2.837 | 56.641 | 3.118/2.416 | 58.2 | 3.118/2.416 | 61.704 | 3.707/2.731 | 36.695 | 3.563/2.231 | 54.444↑ | 3.505/2.528↑ |
| SNR-Aware | 47.025 | 2.971/2.144 | 48.685 | 2.646/1.967 | 49.216 | 2.646/1.967 | 46.441 | 2.938/2.131 | 23.186 | 2.904/1.839 | 42.911 | 2.798/2.005 |
| SNR-Aware with GT-meanloss | 47.43 | 3.067/2.180 | 48.78 | 2.712/1.973 | 49.008 | 2.712/1.973 | 46.602 | 2.956/2.113 | 23.853 | 3.001/1.848 | 43.135↑ | 2.861/2.012↑ |

# E    QUALITATIVE RESULTS

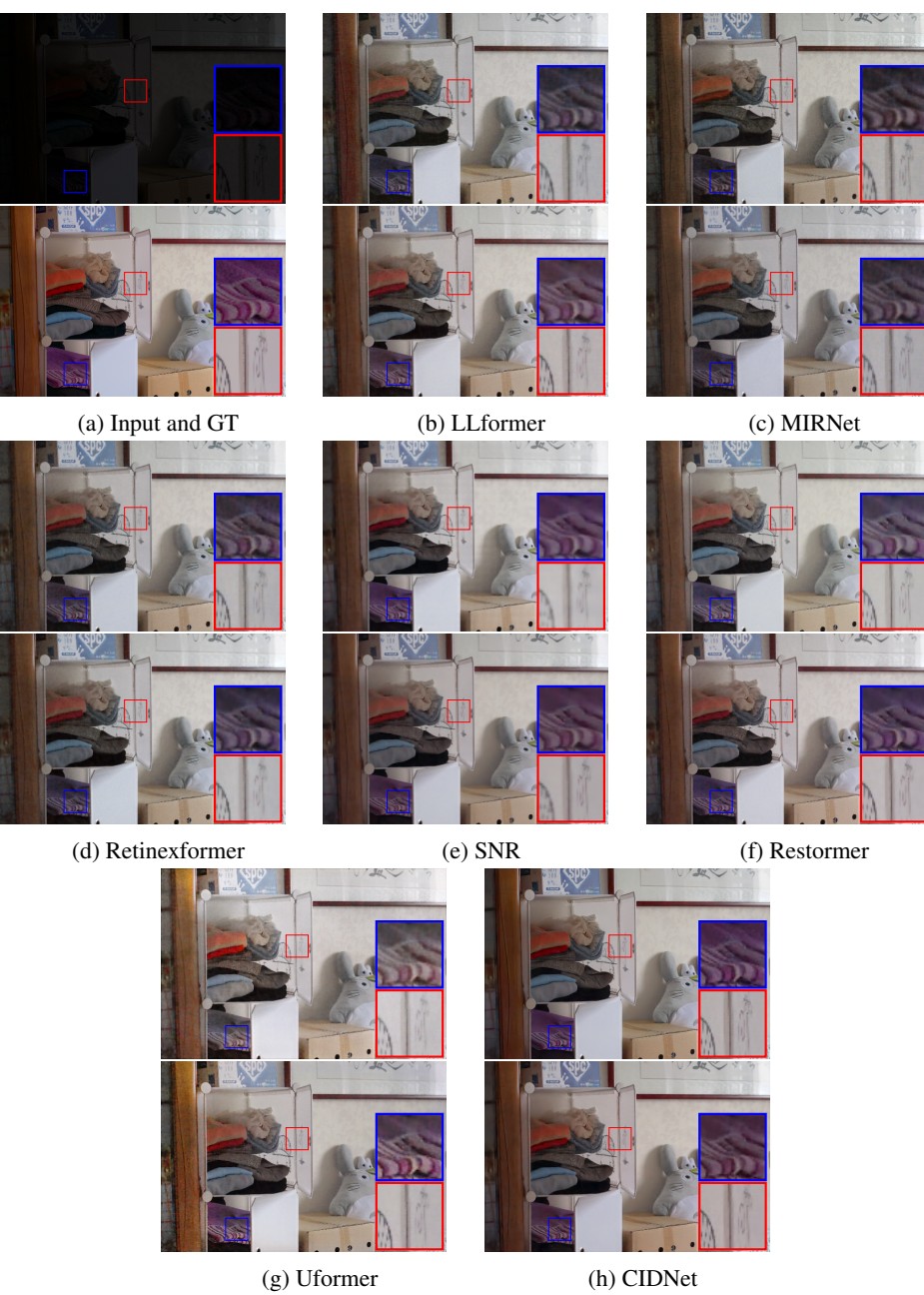

Figure 7: Visual comparison on LOLv1 test dataset. Each set of images is divided into two parts: (a) shows the input image on top and the ground truth (GT) image below; (b)-(h) display the baseline results on top and the corresponding GT-mean loss results below. Overall, the method using GT-mean loss exhibits closer colors and less noise. Additionally, a zoomed-in region is provided for each image to better compare the fine details between the baseline and GT-mean loss-enhanced versions. The methods based on GT-mean loss consistently achieve more accurate colors from (b) to (h). Furthermore, noise is significantly reduced in (b) and (d), and other methods exhibit comparable quality compared to the baseline.

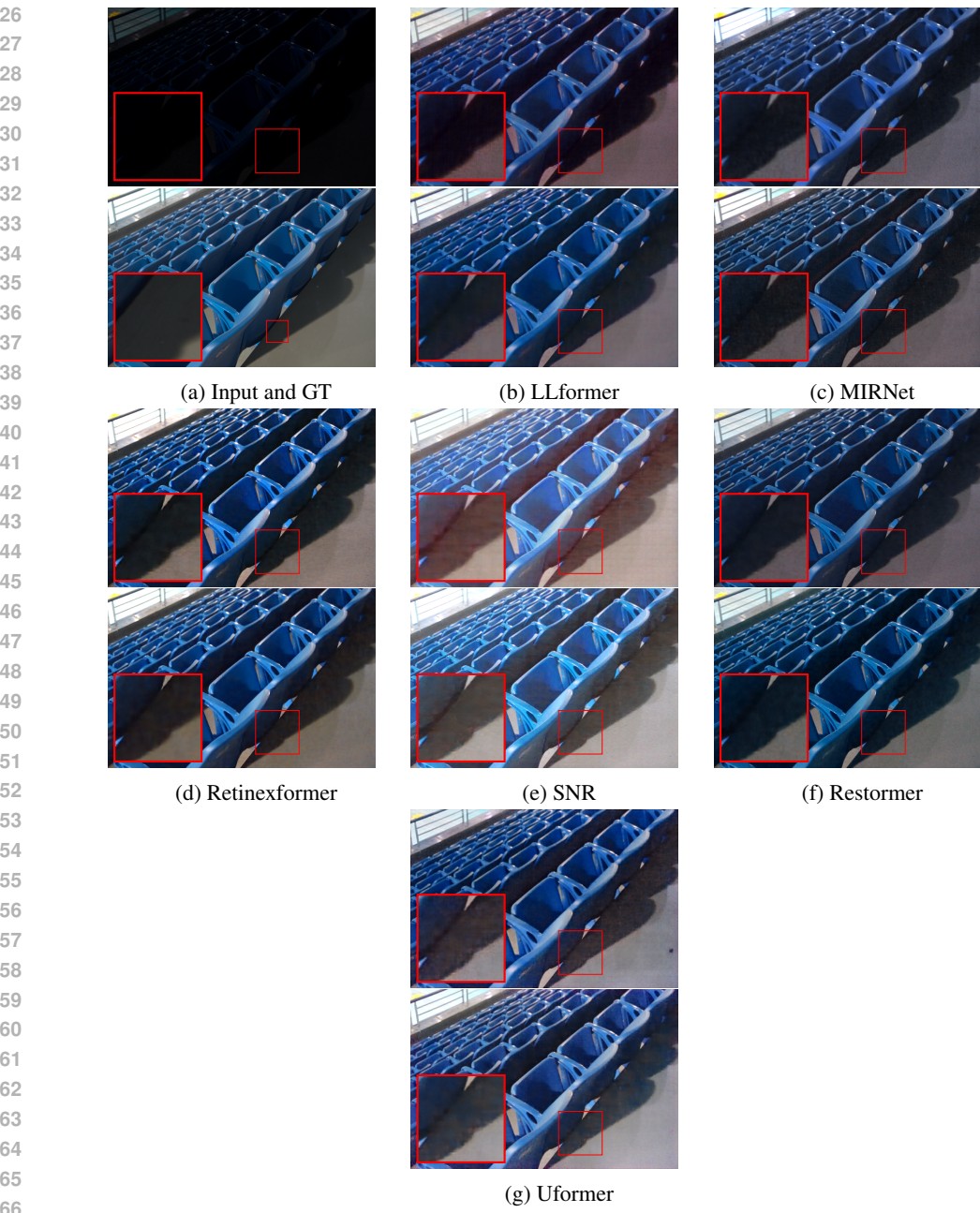

Figure 8: Visual comparison on LOLv2-real test dataset. Each set of images is divided into two parts: (a) display the input image on top and the ground truth (GT) image below; (b)-(g) display the baseline results on top and the corresponding GT-mean loss results below. Overall, the method using GT-mean loss exhibits closer colors and less noise. Additionally, a zoomed-in region is provided for each image to better compare the fine details between the baseline and GT-mean loss-enhanced versions. The methods based on GT-mean loss achieve more accurate colors consistently from (b) to (g). Moreover, noise is significantly reduce in (b), (d), (e), and (g), and other methods exhibit comparable quality to the baseline.

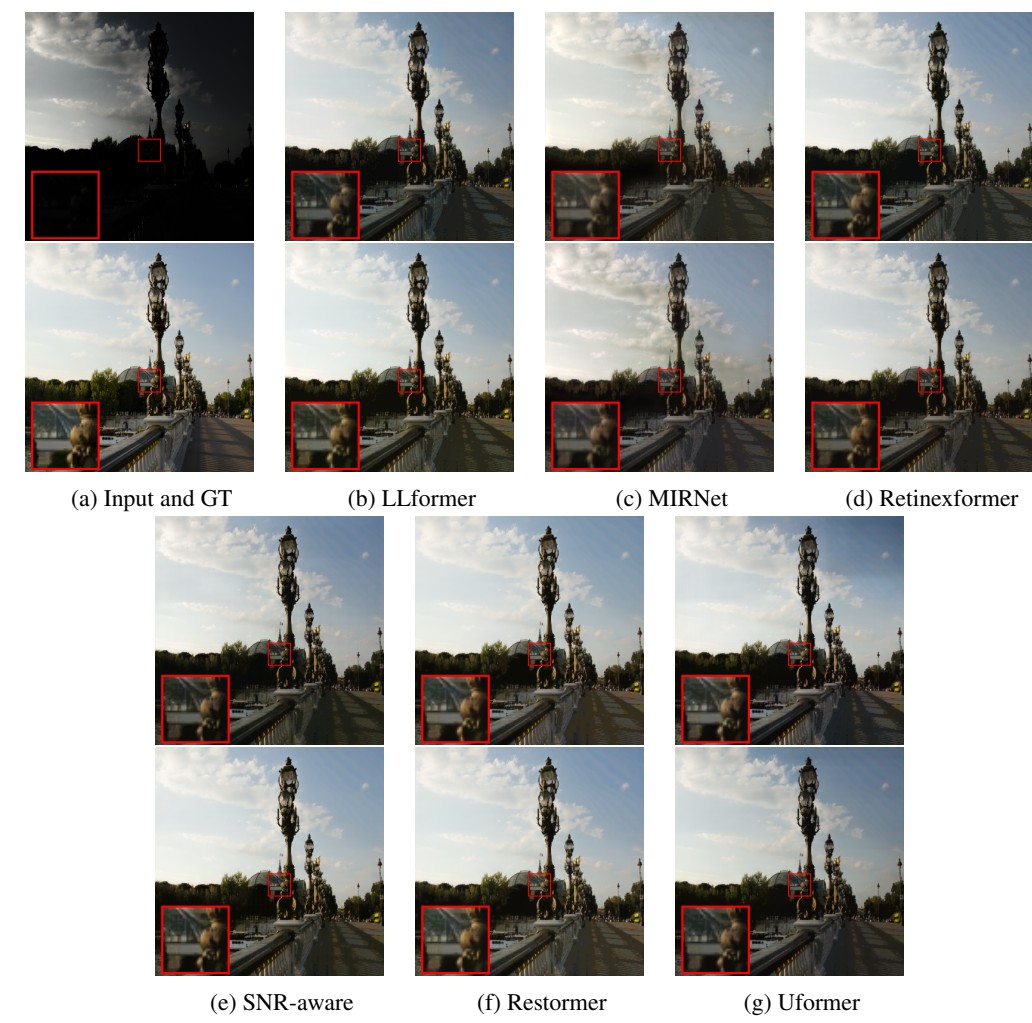

(a) Input and GT        (b) LLformer        (c) MIRNet        (d) Retinexformer

(e) SNR-aware        (f) Restormer        (g) Uformer

Figure 9: Visual comparison on LOLv2-Synthetic test dataset. Each set of images is divided into two parts: (a) shows the input image on top and the ground truth (GT) image below; (b)-(g) display the baseline results on top and the corresponding GT-mean loss results below. Additionally, a zoomed-in region is provided for each image to better compare the fine details between the baseline and GT-mean loss-enhanced versions. The methods based on GT-mean loss achieve more accurate colors, except for (f), where the image quality is also comparable to the original.

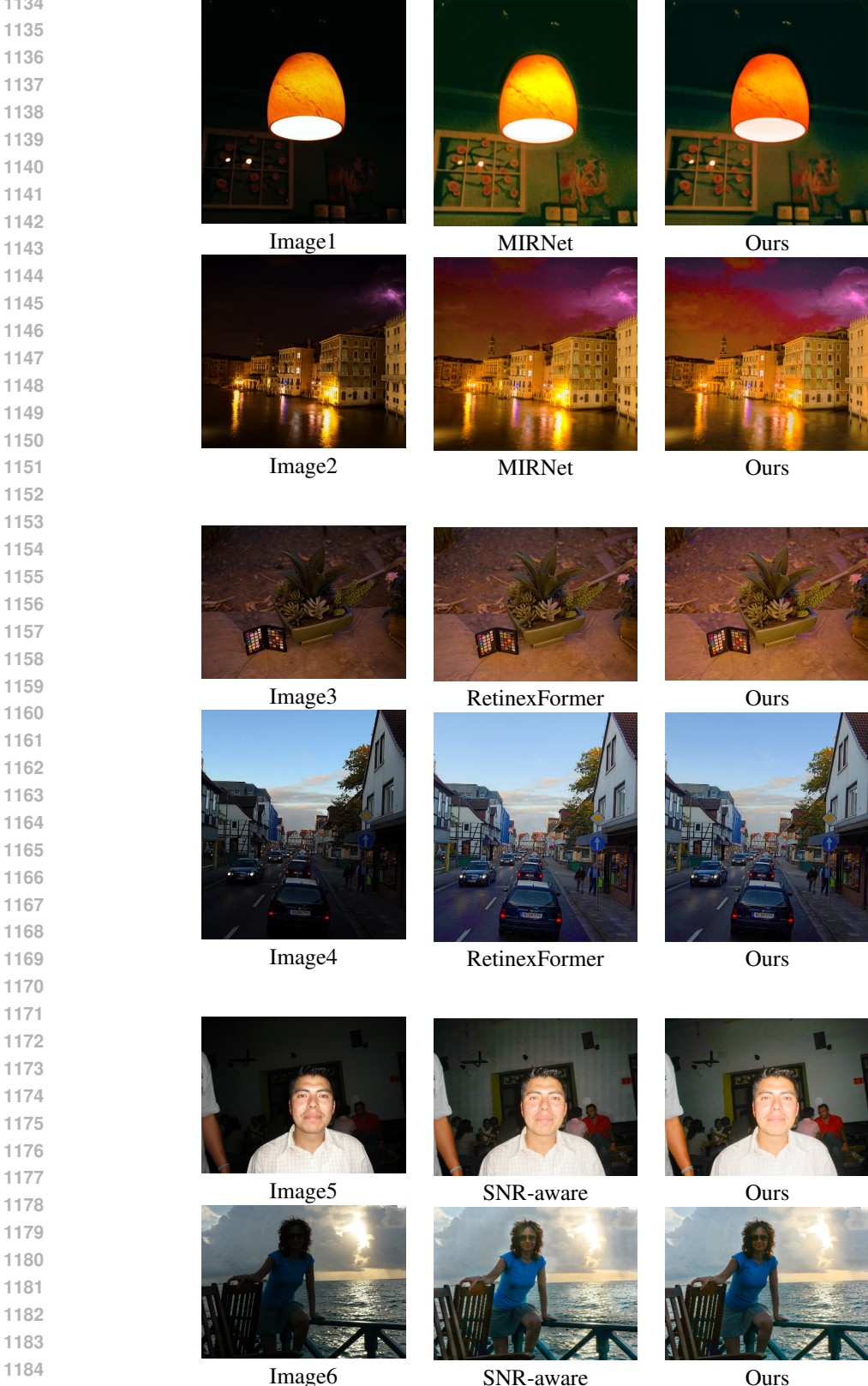

Image1 — MIRNet — Ours

Image2 — MIRNet — Ours

Image3 — RetinexFormer — Ours

Image4 — RetinexFormer — Ours

Image5 — SNR-aware — Ours

Image6 — SNR-aware — Ours

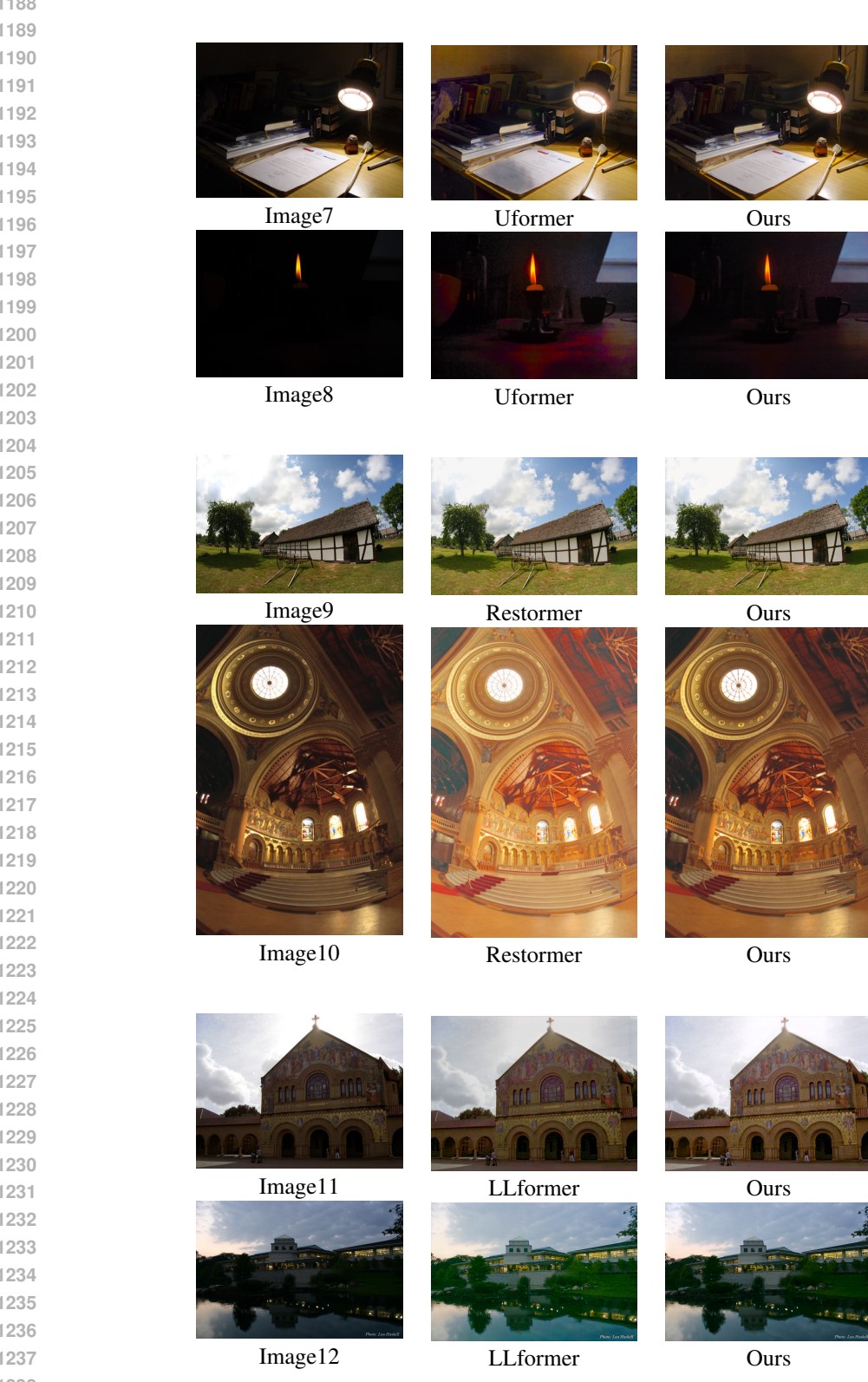

Image7      Uformer      Ours

Image8      Uformer      Ours

Image9      Restormer      Ours

Image10      Restormer      Ours

Image11      LLformer      Ours

Image12      LLformer      Ours

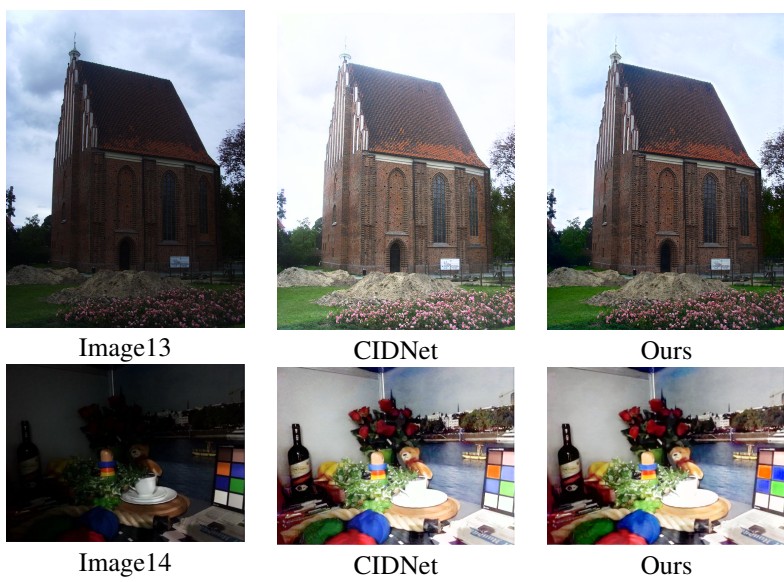

Figure 10: Visual comparison on five unpaired datasets. We selected two image in each baseline to compare. GT-mean loss enhances dark details and color to a suitable interval, which is better than the corresponding baseline. Specifically, the methods based on GT-mean loss achieve better exposure control in images 2-3, 8, and 10-13, enhancing the details in the dark areas to an appropriate level, while the baseline exhibits artifacts and color distortion due to overexposure in these images. In images 4 (The road in the lower left corner), 9(the roof), and 14(the palette), methods based on GT-mean loss provide more accurate colors. Additionally, methods based on GT-mean loss significantly suppress artifacts in images 1 and 5-8.

