# OpenReview forum: "GT-Mean Loss: A Simple Yet Effective Solution for Brightness Mismatch in Low-Light Image Enhancement"
_ICLR.cc/2025/Conference — ICLR 2025 Conference Withdrawn Submission_

### Official Review · Reviewer_wA78 · 2024-10-30

**Soundness:** 3
**Presentation:** 3
**Contribution:** 1
**Rating:** 3
**Confidence:** 5

**Summary:**

This paper proposes a simple and effective GT-mean loss for low-light image enhancement. The GT-mean loss aims to align the average image pixel value intensities of the enhanced image to that of GT image. The authors provide an analysis from the point of view of probabilistic modeling. A design of the balancing weights between L1 loss and GT-enhance loss based on KL divergence is designed.

**Strengths:**

(a) The presentation of this paper is well-dressed. The writing is good and easy-to-follow. Although the proposed method is simple, but I found some insights in the weight design part.

(b) The experimental results are good and solid. The authors have validated the effectiveness of the proposed GT-mean loss on seven methods on three datasets in both terms of normal enhanced results and GT-mean enhanced results. The Visual comparison also looks good. The ablation studies can demonstrate the effectiveness of the proposed techniques.

**Weaknesses:**

(a) The novelty is poor. The GT-mean enhancement is actually a cheat. It was first used by KinD [1]. However, KinD does not achieve very high results. Thus, this trick does not attract much attention. Later work LLflow [2] uses this trick to achieve a very high result on LOL-v1 dataset. Since then on, more and more methods used this trick to pursue better PSNR and SSIM. Although Retinexformer pointed out this is wrong and bad, it cannot stop the community from using this trick. From my point of view, using the average value of GT image pixels to achieve higher numeric results does not make sense. It is not essential. And this is just a small trick. I do not think this trick is worth an ICLR publication.

[1] Kindling the Darkness: A Practical Low-light Image Enhancer. ACM MM 2019.

[2] Low-Light Image Enhancement with Normalizing Flow. AAAI 2022.

(b) The proposed GT-Mean Loss does not work in any case, especially for the out-of-distribution scenes. For example, in lines 1123 - 1129,
the original LLformer performs better without the GT-Mean Loss. This is because the correction of GT-Mean has a bias on the trained dataset. This is the main technical drawback of the proposed method.

(c) Algorithm 1 is unnecessary since it is very simple. There is no need to draw the algorithm table.

(d) Code and pre-trained weights are not submitted, the reproducibility cannot be checked.

**Questions:**

The proof in Appendix A is just the normal definition and formulation of the KL divergence. Why write it? it seems unnecessary.

---

> ### Author Response · Authors · 2024-11-21
>
> Thank you for reading, we respond to them one by one.
>
>
>
> -  W (a):
>      We acknowledge that there are different opinions on the use of GT-Mean metrics in LLIE community.  Some researchers oppose this technique, arguing that GT-Mean metrics involve post-processing that may lead to ground truth (GT) information leakage.
>
>      However, **our proposed GT-Mean Loss does not share the same concerns as GT-Mean metrics.** Unlike GT-Mean metrics, GT-Mean Loss does not perform GT-Mean adjustment on the model’s initial input. Instead, it applies GT-Mean adjustment only during the computation of the loss function. We kindly ask you to **consider the fundamental differences**.
>
>     Finally, we argue that GT-Mean metrics have merit because they allow comparisons of image quality **at a same brightness level**, eliminating the impact of brightness mismatch on evaluation metrics. One of the goals of our paper is to advocate for the LLIE community to report **both standard metrics and GT-Mean metrics simultaneously** to enable a more comprehensive evaluation of low-light image quality.
>
> -  W (b): About the Generalization of GT-Mean Loss to out-of-distribution scenes.
>     To address this concern, we now include supervised dataset metrics like LPIPS and QALIGN and unsupervised dataset metrics like MUSIQ and QALIGN in Appendix D (or see in our comments for reviewer 5 DCR). The results based on new metrics on unpaired datasets show that our improvements also apply to the out of distribution case, not due to the bias on the training dataset.
>
>     Additionally, we provide detailed metrics for the images discussed in lines 1123-1129:
>
>     - **For LLFormer (baseline):**
>
>         - MUSIQ $\uparrow$: 64.073
>
>         - QALIGN $\uparrow$: 4.078125 (IQA) / 2.885 (IAA)
>
>     - **For LLFormer with GT-Mean Loss (ours):**
>
>         - MUSIQ $\uparrow$: 65.889
>
>         - QALIGN $\uparrow$: 4.145 (IQA) / 2.898 (IAA)
>
>     Our method achieves better results than the baseline across all metrics (higher MUSIQ and QALIGN scores indicate better performance).
>
>     From a visual perspective, the baseline output exhibits color distortion, with a slight greenish tint on the walls, while our method produces more natural results.
>
>
>
> - W (C) and Q:
>
>     Thank you for the suggestion. We have removed Algorithm 1 to make the paper more concise and added a more in-depth analysis of $W$ in Appendix A.
>
> - W (D):
>
>     We have now provided the code and checkpoints. [https://anonymous.4open.science/r/Retinexformer-FEC9/](https://anonymous.4open.science/r/Retinexformer-FEC9/)
>
> *Looking for your feedback*

---

### Official Review · Reviewer_rJj2 · 2024-11-02

**Soundness:** 2
**Presentation:** 3
**Contribution:** 2
**Rating:** 5
**Confidence:** 5

**Summary:**

Low-light image enhancement (LLIE) aims to improve the visual quality of images captured under poor lighting conditions. In supervised LLIE tasks, there exists a significant yet often overlooked inconsistency between the overall brightness of an enhanced image and its ground truth counterpart, referred to as brightness mismatch in this study. Brightness mismatch negatively impact supervised LLIE models by misleading model training. However, this issue is largely neglected in current research. In this context, we propose the GT-mean loss, a simple yet effective loss function directly modeling the mean values of images from a probabilistic perspective. The GT-mean loss is flexible, as it extends existing supervised LLIE loss functions into the GT-mean form with minimal additional computational costs. Extensive experiments demonstrate that the incorporation of the GT-mean loss results in consistent performance improvements across various methods and datasets.

**Strengths:**

The paper propose GT-mean loss, a simple yet effective loss function directly modeling the mean values of images from a probabilistic perspective.

**Weaknesses:**

* The authors in this paper define a new term, “brightness mismatch” which refers to the “inconsistency” between enhanced image and the ground-truth image (lines  028-030). However, it is quite unclear as to how it impacts the model training as mentioned in lines 030-032. The authors are trying to imply that the outcome of the trained model affects the training of the model? It is somewhat difficult to follow. The same has several occurrences in the manuscript, but it is not quite clear how the brightness mismatch is affecting the model training. The authors are requested to provide clear and crisp information regarding the claims made.
* Figure 2 and the explanation provided is somewhat misleading/not easy to understand. The “scaled” version of the GT image supposedly has “infinite” GT-mean PSNR Values; this seems quite untrue. This brings to next discrepancy/confusion in the paper, the GT-mean PSNR. The manuscript seems to not have a said definition for GT-mean PSNR. The authors are requested to provide more information and clarity on this. There is no information as to how the GT-mean PSNR is computed to recreate the Figure 2, and also how the result of infinity is achieved. Is it for some kind of demonstration purpose?
* Lines 086-094 are not quite clear. Several Methods in literature make use of non-PSNR like metrics such as perceptual loss [1], color fidelity losses [2] which somewhat aim at solving the “brightness mismatch” issue as claimed by the authors. Towards this the authors are requested to provide adequate information for the claims made.
* Lines 230-234 are extremely difficult to follow. For example., what is the meaning of “At this stage, the GT-mean loss value measures the fidelity between the two images by excluding brightness mismatch in advance considering brightness mismatch, thereby sustaining effective model training”? The writing style and the information provided at this point makes it quite difficult to comprehend the manuscript.
* The overall contribution of this paper is supposedly GT-mean “loss” but the authors (in lines 122-125) describe it as “extension” to any existing loss function pushing the reader in the direction of if this is really a loss function or rather a regularization term.
* The manuscript overall is rather difficult to follow and understand.
* The qualitative results in section E show least/minimal differences between the baseline results and the proposed GT-mean version. There is no drastic change in the results shown and supposedly a non-issue due to the availability of losses like perceptual loss, color loss, cosine similarity loss for minimizing color deviations. The observed improvements seem to not fit for the level to claims made.



References
[1] Desai, Chaitra, et al. "Lightnet: Generative model for enhancement of low-light images." Proceedings of the IEEE/CVF international conference on computer vision. 2023.
[2] Ignatov, Andrey, et al. "Dslr-quality photos on mobile devices with deep convolutional networks." Proceedings of the IEEE international conference on computer vision. 2017.

**Questions:**

Authors to consider major concerns mentioned.

---

> ### Author Response · Authors · 2024-11-21
>
> Thanks for your comments and suggestions. We respond to them one by one.
>
> - W 1:
>
> > it is unclear as to how brightness mismatch impacts the model training as mentioned
>
> In fact, we demonstrated in the 'Impact on Training' section of the Introduction that the loss value due to brightness mismatch does not match the actual image quality, which affects the training process. Additionally, in Section 4.3 and Figures 4 (a) and 4 (b), we provided the actual training curves of the model, showing the influence of GTmeanloss on the training process.
>
> - W 2:
>
> Thank you for your suggestion. We have revised the expression to improve the understanding of the article.
> In lines 99-101, we mentioned the definition of the GTmean metric to align with the reading sequence. Now, we have included the definition of the GTmean metric in the caption of Figure 2.
>
> **Specific changes:**
> *For the scaled image, GT-mean PSNR is computed as $PSNR(\frac{E[GT]}{E[GT\times0.8]}GT\times0.8,GT)$, resulting in an infinite PSNR value (In practice, PSNR usually has a pre-defined upper bound. In this figure, we strictly follow the mathametical definition of PSNR for demonstration purposes).*
>
> -  W 3:
>
> It is worth noting that perceptual loss (or color fidelity losses) typically requires joint use with pixel-level loss (L1/MSE-like) to achieve pixel-level predictions for images. Using these perceptual losses alone is challenging for pixel-level predictions. However, pixel-level loss, due to its lack of perceptual capabilities, is affected by brightness mismatch.
>
> In our experiments, we chose SNR and CIDnet as baselines, both of which use perceptual loss (as shown in Table 2). After extending their losses to GTmean loss, we observed a significant performance improvement. This demonstrates that even after using perceptual loss, it is difficult to eliminate the negative impacts caused by incorrect pixel-level loss. GTmean loss, however, eliminates this impact and achieves performance improvement.
>
>
> - W 4:
>
> Thank you for your suggestion. We have rephrased the sentence as follows: In this stage, the GT mean loss primarily compares image differences at the same mean brightness to avoid the negative effect of brightness mismatch, thereby maintaining effective model training.
>
> - W 5:
>
> The specific form of the GT-mean loss depends on the original loss  $L(\cdot)$, as shown in Eq 1. We only require that the original loss is a supervised loss (with the input image and corresponding ground truth), and there are no additional constraints on $L(\cdot)$. For example, the L1 loss can be extended to the corresponding GT-mean L1 loss.
>
> -  W 7：
>
> Since we did not modify any structures of the original baseline, there are no significant visual differences in supervised datasets, with improvements mainly observed in statistical metrics. Our method shows significant visual improvements in unsupervised datasets, demonstrating the good generalization of GT-mean loss. Additionally, we now provide more metrics to prove the effectiveness of our method (see in  **Appendix D** or our comments for  Reviewer **5DCR**) and offer an anonymous code repository for quick reproduction of GT-mean loss. Welcome to try GT-mean loss in https://anonymous.4open.science/r/Retinexformer-FEC9/
>
> *Looking forward to your feedback*

---

### Official Review · Reviewer_LK8R · 2024-11-03

**Soundness:** 3
**Presentation:** 2
**Contribution:** 3
**Rating:** 5
**Confidence:** 5

**Summary:**

The authors propose to align overall brightness of the enhanced image with ground-truth images for supervision in low-light enhancement models. Experiments have shown that the proposed GT-mean loss results in consistent performance gain in PSNR.

**Strengths:**

1. The proposed method is well-motivated.  The MSE/L1-based metrics, say PSNR or RMSE, do have a problem identifing noise from lightness bias in RGB color space.
2. The experiment results are reasonable.

**Weaknesses:**

1. Novelty. The GT-mean metric was first proposed in KinD[1], and has been adopted by many previous methods[2,3]. I don't think the effectiveness of GT-mean **metric**, can be considered as a contribution.
2. Formulation of global lightness. The author models the global brightness with normal distribution, which breaks in most real-world scenarios.

[1] Zhang et. al, Kindling the Darkness: A Practical Low-light Image Enhancer, ACM MM 2019.

[2] Wang et. al, Low-Light Image Enhancement with Normalizing Flow, AAAI 2022.

[3]Cui et. al, Retinexformer: One-stage Retinex-based Transformer for Low-light Image Enhancement, ICCV 2023.

**Questions:**

1.  Estimating W with the average of two KL divergences is confusing. Why? More in-depth analysis should be conducted.
2.  In Figure 3, the authors plotted average brightness. Is this average brightness distribution obtained from images? In my experiments, the brightness distribution of images (widely visualized as a histogram) doesn't always come with a perfect Gaussian distribution.
3. With the proposed gt mean loss, the models are still aligning input with reference images. However, as an **enhancement** task, there exists no ground truth image. It is not surprising that gt-mean alignment helps align the output with reference images, and thus improves the performance on reference-based metrics, say, PSNR and SSIM. Please report the non-reference metrics MUSIQ[1] and Q-align[2] for DICE, MEF, LIME, NPE, and VV datasets.

[1]Ke et. al,  MUSIQ: Multi-Scale Image Quality Transformer, ICCV 2021.

[2] Wu et. al, Q-Align: Teaching LMMs for Visual Scoring via Discrete Text-Defined Levels, ICML 2024.

---

> ### Author Response · Authors · 2024-11-21
>
> Thanks for your comments and suggestions. We respond to them one by one.
>
> - W1:
>
> The low-light enhancement (LLIE) community currently holds divided opinions on the GT-mean metric. Some researchers argue that it is merely a technique unworthy of broader adoption (a point of contention we share with reviewer **wA78**).
>
> In this paper, we demonstrate the **effectiveness of the GT-mean metric** and highlight its value as a robust supplement to existing reference-based evaluation metrics. As part of our contribution, we advocate for the LLIE community to _simultaneously report_ both conventional metrics and GT-mean metrics. Reporting results based on both metric types ensures a more comprehensive evaluation of low-light image quality and addresses the concerns surrounding fairness and reproducibility in LLIE benchmarking.
>
> We hope this perspective aligns with the broader interests of the LLIE community. Thank you for raising this important discussion!
>
> - W2 and Q 2:
>
> The average brightness distribution is not derived from the statistics of multiple images but rather represents an assumption about the mean brightness of a **single image** . We propose that a single image may exhibit varying levels of average brightness, and the probabilities of these potential average brightness levels follow a Gaussian distribution.
>
> - Q 1:
>
> Thank you for your suggestion, which has significantly improved the quality of our paper. The design of $W$ was inspired by the construction of an intermediate distribution in JSD, which involves calculating KL divergence twice. However, since JSD does not have an analytical form, it cannot be computed efficiently for $W$.
>
> To address this, we designed $W$ in a way that provides an analytical form, enabling faster computation. Additionally, we discovered a connection between $W$ and the Bhattacharyya distance, which we discuss in detail in **Appendix A**.
>
> For convenience, the connection between $W$ and the Bhattacharyya distance is as follows:
>
> Let $p (x)=\mathcal{N}(\mu_1, \sigma_1)$ and $q (x)= \mathcal{N}(\mu_2, \sigma_2)$. According Eq. 5, we have: $$ W = \frac{1}{2} \left[ \log \frac{\sigma_m}{\sigma_1} + \frac{\sigma_1^2 + (\mu_1 - \mu_m)^2}{2 \sigma_m^2} - \frac{1}{2} \right] + \frac{1}{2} \left[ \log \frac{\sigma_m}{\sigma_{2}} + \frac{\sigma_{2}^2 + (\mu_{2} - \mu_m)^2}{2 \sigma_m^2} - \frac{1}{2} \right]. (8)
> $$ where $\mu_m = \frac{\mu_{1}+\mu_{2}}{2}$, $\sigma_m^2 = \frac{\sigma_{1}^2+\sigma_{2}^2}{2}$. We proof Eq. 8 can be written as $$W=\frac{1}{4}\frac{(\mu_1-\mu_2)^2}{\sigma_1^2+\sigma_2^2}+\frac{1}{2}\log\left(\frac{\sigma_1^2+\sigma_2^2}{2\sigma_1\sigma_2}\right), (9) $$ where Eq. 9 is also the closed form of the Bhattacharyya distance after two one-dimensional Gaussian distributions [1].
>
> *Proof.* We can extend the first term in as:
> $$
> \frac12\left[\log\frac{\sigma_m}{\sigma_1}+\frac{\sigma_1^2+(\mu_1-\mu_m)^2}{2\sigma_m^2}-\frac12\right] =\frac12\log\frac{\sigma_m}{\sigma_1}+\frac12\cdot\frac{\sigma_1^2+(\mu_1-\mu_m)^2}{2\sigma_m^2}-\frac14. (10)
> $$ Similarly, the second term can be extended as:
> $$
> \frac12\left[\log\frac{\sigma_m}{\sigma_{2}}+\frac{\sigma_{2}^2+(\mu_{2}-\mu_m)^2}{2\sigma_m^2}-\frac12\right] =\frac12\log\frac{\sigma_m}{\sigma_{2}}+\frac12\cdot\frac{\sigma_{2}^2+(\mu_{2}-\mu_m)^2}{2\sigma_m^2}-\frac14. (11)
> $$ Combine this two term, we have:
> $$
> \frac12\log\frac{\sigma_m^2}{\sigma_1\sigma_2}+\frac12\cdot\frac{\sigma_{1}^2+\sigma_{2}^2+(\mu_{1}-\mu_m)^2+(\mu_{2}-\mu_m)^2}{2\sigma_m^2}-\frac12.(12)
> $$ Let $p (x)=\mathcal{N}(\mu_1, \sigma_1)$ and $q (x)= \mathcal{N}(\mu_2, \sigma_2)$.
> Due to $\mu_m = \frac{\mu_{1}+\mu_{2}}{2}$, $\sigma_m^2 = \frac{\sigma_{1}^2+\sigma_{2}^2}{2}$, Eq. 12 can be written as:
> 	$$
> 		\frac 12\log\frac{\sigma_m^2}{\sigma_1\sigma_2}+\frac 12\cdot\frac{\sigma_{1}^2+\sigma_{2}^2+ \frac{(\mu_{1}-\mu_{2})^2}{2}}{2\sigma_m^2}-\frac 12\\
> 		$$
> 		$$
> 		=\frac 12\log\frac{\sigma_{1}^2+\sigma_{2}^2}{2\sigma_1\sigma_2}+\frac 12\cdot\frac{\sigma_{1}^2+\sigma_{2}^2+ \frac{(\mu_{1}-\mu_{2})^2}{2}}{\sigma_{1}^2+\sigma_{2}^2}-\frac12
> 		$$
> 		$$=\frac 14\frac{ (\mu_{1}-\mu_{2})^2}{\sigma_{1}^2+\sigma_{2}^2}+\frac 12\log\frac{\sigma_{1}^2+\sigma_{2}^2}{2\sigma_1\sigma_2}$$
> 	This concludes the proof.
>
> [1] Ravi Kashyap,The perfect marriage and much more: Combining dimension reduction, distance measures and covariance, Physica A: Statistical Mechanics and its Applications, 2019.
>
>
> - Q3:
>
>
> Thank you for your valuable suggestions. We have provided additional quantitative results in **Appendix D** (or refer to our comments for reviewer **5DCR**). For datasets such as LOLv1 and LOLv2, we report LPIPS and Q-ALIGN (IQA\IAA). For no-reference datasets like DICE, MEF, LIME, NPE, and VV, we provide MUSIQ and Q-ALIGN (IQA\IAA).
>
> The newly added metrics demonstrate that our method consistently outperforms the baseline across various benchmarks, further validating its effectiveness in enhancing image quality.
>
> *Looking for your feedback*

---

### Official Review · Reviewer_25iy · 2024-11-03

**Soundness:** 2
**Presentation:** 3
**Contribution:** 2
**Rating:** 5
**Confidence:** 4

**Summary:**

This paper proposes a GT-mean loss to address the brightness mismatch in low-light image enhancement. The authors first define the brightness mismatch and describe its impact on evaluation and training. Subsequently, the GT-mean loss and weight design are introduced. The evaluation is performed on several backbones with three low-light enhancement datasets. The GT-mean loss can consistently improve performance.

**Strengths:**

1.	The proposed loss leads to consistent performance for several backbones on three low-light image enhancement datasets.
2.	The paper is well-motivated and elaborately develops a loss function for the observation.

**Weaknesses:**

1.	In Fig.2, it seems more like a metric problem. If PSNR is not sensitive to this mismatch, why can the proposed loss function improve the PSNR performance? When there is a resolution difference between the original GT and the scaled image, how do we compute the PSNR value of the scaled one? Moreover, how is the brightness mismatch problem illustrated in Fig.2?
2.	The paper couples the original loss and the introduced one using W. How do we confirm that these two loss functions have any relationships?
3.	Tab. 1 shows that perceptual loss can also identify the scaled image as better. Can this loss make a more significant improvement than the proposed one when applied to algorithms that initially did not use this loss?
4.	Some typos. Lv1->LOLv1. The citation for ZeroDCE is missing.

**Questions:**

Please see weaknesses

---

> ### Author Response · Authors · 2024-11-21
>
> Thanks for your comments and suggestions. We respond to them one by one.
>
> - W 1:
>
> > In Fig. 2, it seems more like a metric problem. If PSNR is not sensitive to this mismatch, why can the proposed loss function improve the PSNR performance?
>
> The GT-Mean Loss addresses image degradation from two perspectives: The first term ensures that the output image $f(x)$ closely matches the ground truth (GT). The second term ensures that  $\lambda f(x)$  (where $\lambda$ is a scalar) closely matches the GT after brightness adjustment.
>
> Since $\lambda f(x)$  is a linear combination of $f(x)$  , the optimal values for both loss terms occur when $\lambda =1$ and $f(x)=GT$. Even though PSNR may not directly capture brightness mismatch, the minimization from these two aspects ensures a better overall match between the output and the ground truth, which ultimately improves PSNR.
>
> >  When there is a resolution difference between the original GT and the scaled image, how do we compute the PSNR value of the scaled one?
>
> The scaled image in our experiments refers to an image with its brightness scaled by 0.8, not its resolution. To avoid confusion, we have revised the caption for Fig. 2 as follows:
>
> **Specific changes:**
> *the original image's brightness scaled by a factor of 0.8*
>
> >  Moreover, how is the brightness mismatch problem illustrated in Fig. 2?
>
> The PSNR of the noisy image is higher than that of the scaled image, but this does not align with the actual image quality. This discrepancy occurs because the brightness difference between the scaled image and the ground truth (GT) is too large (brightness mismatch), causing PSNR to fail to capture the true image quality.
>
> - W 2:
>
> >  The paper couples the original loss and the introduced one using W. How do we confirm that these two loss functions have any relationships?
>
> The first term of the GT-Mean Loss is the original loss applied to the model's output. The second term calculates the original loss again after adjusting the model's output via GT-Mean. The weighting factor $W$ dynamically determines the importance of each term in training process.
>
> Effectively, the GT-Mean Loss calculates the original loss twice (once for the raw output and once for its GT-Mean adjustment), ensuring the preservation of the original loss's functionality while mitigating the effects of brightness mismatch.
>
>
>
> - W 3:
>
> > _Tab. 1 shows that perceptual loss can also identify the scaled image as better. Can this loss make a more significant improvement than the proposed one when applied to algorithms that initially did not use this loss?_
>
> It is important to note that our method and perceptual loss can be used together to achieve even better performance. In our baseline experiments (e.g., SNR and CIDNET), perceptual loss is already included. By extending these baselines with GT-Mean Loss, we achieved further improvements. This demonstrates that GT-Mean Loss is complementary to perceptual loss and can enhance performance when used together.
>
> - W 4:
>
> > _Some typos. Lv1 -> LOLv1. The citation for ZeroDCE is missing._
>
> Thank you for carefully reviewing the manuscript. We have corrected the typos and added the missing citation for ZeroDCE in the revised version.
>
>
> *Looking for your feedback*

---

### Official Review · Reviewer_5DCR · 2024-11-06

**Soundness:** 2
**Presentation:** 2
**Contribution:** 3
**Rating:** 5
**Confidence:** 4

**Summary:**

The paper introduces GT-Mean Loss, a novel loss function designed to address the issue of brightness mismatch in low-light image enhancement (LLIE). GT-Mean Loss aims to align the brightness of enhanced images with their ground truth counterparts by dynamically adjusting the loss during training, ensuring that the model focuses on factors beyond brightness mismatches.

**Strengths:**

1. Proposes a straightforward loss function that effectively mitigates brightness mismatches in LLIE.
2. Demonstrates performance improvements across various LLIE models, supporting the generalizability of GT-Mean Loss.
3. The approach adds minimal computational overhead, making it easy to integrate into existing models without significant resource costs.

**Weaknesses:**

The authors claim that "brightness mismatch dominates the PSNR values," suggesting that brightness inconsistency heavily biases this commonly used metric, leading to inaccurate quality evaluations. However, this assertion may oversimplify the limitations of PSNR, as the issue presented seems to be a specific type of counterexample rather than an overarching flaw in PSNR itself. PSNR is limited primarily in its sensitivity to perceptual qualities rather than simply brightness mismatch, and as such, is inherently less reliable for subjective quality evaluation. It’s possible to generate similar counterexamples through methods like the MAD competition, which reveal PSNR’s broader limitations in capturing perceptual quality accurately. Further, rather than focusing only on brightness mismatch, utilizing perceptually-oriented image quality assessment (IQA) metrics, such as LPIPS and DISTS, would provide a more holistic and accurate quality differentiation. These metrics are specifically designed to capture perceptual differences that metrics like PSNR or SSIM may overlook. By comparing GT-Mean Loss to metrics like LPIPS and DISTS, the authors could strengthen the argument for their loss function's contribution to enhancing perceptual quality in LLIE tasks, rather than solely addressing the limited case of brightness mismatch.

**Questions:**

1. Although the paper presents GT-Mean PSNR and GT-Mean SSIM as enhanced metrics, the evaluation lacks comparison using perceptual quality metrics such as LPIPS, DISTS, Q-Align, and LIQE, which could better reflect subjective image quality.

2. The method’s focus on brightness mismatch as the sole training issue may overlook other complex image degradation factors in low-light settings, such as the noise, the color bias, the unaccurate white balance,  limiting the applicability of the method for more holistic quality enhancement.

3. The performance improvements presented in the paper, while consistent, appear modest in scale. The GT-Mean Loss achieves incremental gains in metrics like PSNR and SSIM, but these enhancements are relatively small and may not justify the added complexity of implementing a new loss function focused on brightness alignment alone. For real-world applications, such minor improvements could be seen as insufficient.

4. In addition to quantitative metrics, conducting a user study on the enhanced images could provide a more reliable and insightful demonstration of GT-Mean Loss’s impact on perceptual quality. Objective metrics, especially pixel-based ones like PSNR and SSIM, do not fully capture human perception

5. The captions in the paper’s figures would benefit from additional context, allowing them to convey key insights independently of the main text. Currently, they lack sufficient detail to stand alone, which can make it challenging to grasp the full significance of the visual data without referring back to the text.

6. The motivation section, particularly the second paragraph of the Introduction, would benefit from a clearer and more robust explanation. Currently, the rationale behind the proposed GT-Mean Loss is somewhat limited.

---

> ### Author Response · Authors · 2024-11-21
>
> Thank you for reading, your insights on image quality evaluation have enhanced the quality of our articles and now we are responding to your queries one by one.
>
> - W and Q 1:
> 	Thank you for your valuable suggestion. In **Appendix D**, we provide additional quantitative results. For datasets like LOLv 1 and LOLv 2, we report LPIPS and Q-ALIGN (both IQA and IAA). For datasets without reference images, such as DICE, MEF, LIME, NPE, and VV, we report MUSIQ and Q-ALIGN (IQA and IAA). The new metrics consistently demonstrate that our method outperforms the baseline, further validating its effectiveness in improving image quality.
>
> 	For convenience, the results based on new metrics are as follows:

---

> ### Author Response · Authors · 2024-11-21
>
> GT-Mean Loss outperforms in **almost all** metrics on paired datasets (except for ~~strikethrough~~).
>
>
> |                                |          LOLv 1           |                            |                    |        LOLv 2-real        |                            |                    |     LOLv 2-synthetic      |                            |                    |
> | :----------------------------: | :-----------------------: | :------------------------: | :----------------: | :-----------------------: | :------------------------: | :----------------: | :-----------------------: | :------------------------: | :----------------: |
> |                                | Normal Lpips $\downarrow$ | GT-mean Lpips $\downarrow$ | Q-Align $\uparrow$ | Normal Lpips $\downarrow$ | GT-mean Lpips $\downarrow$ | Q-Align $\uparrow$ | Normal Lpips $\downarrow$ | GT-mean Lpips $\downarrow$ | Q-Align $\uparrow$ |
> |         RetinexFormer          |           0.141           |           0.134            |    3.317/1.959     |           0.163           |           0.152            |    3.478/2.009     |           0.064           |           0.057            |    3.148/2.114     |
> | RetinexFormer with GT-meanloss |         **0.138**             |           **0.132**            |    **3.331/1.971**     |          **0.143**          |           **0.134**            |    **3.778/2.048**     |           **0.063**           |           **0.056**            |    **3.191/2.144**     |
> |             MIRNet             |           0.222           |           0.216            |    2.917/1.745     |           0.313           |           0.303            |    2.598/1.520     |           0.122           |           0.114            |    2.956/2.145     |
> |    MIRNet with GT-meanloss     |           **0.196**           |           **0.189**            |    **3.039/1.758**     |           **0.214**          |           **0.208**            |    **2.924/1.702**     |           **0.104**           |           **0.094**            |    **3.064/2.187**     |
> |            LLFormer            |           0.183           |           0.178            |    3.027/1.800     |           0.248           |           0.236            |    2.714/1.590     |           0.07            |           0.064            |    3.102/2.099     |
> |   LLFormer with GT-meanloss    |          **0.138**           |           **0.133**           |    **3.373/1.956**    |           **0.166**          |           **0.156**            |    **3.206/1.884**    |           **0.058**           |           **0.051**            |    **3.197/2.130**    |
> |           Restormer            |           0.128           |           0.122            |    3.567/2.032     |           0.162           |           0.147            |    3.478/1.987     |           0.045           |           0.039            |    3.350/2.187     |
> |   Restormer with GT-meanloss   |           **0.122**           |          **0.117**            |    **3.672/2.054**     |           **0.149**           |           **0.135**            |    **3.554/2.020**     |           **0.041**           |           **0.036**            |    **3.404/2.218**     |
> |            Uformer             |           0.212           |           0.195            |    3.087/1.946     |           0.228           |           0.199            |    2.882/1.827     |           0.06            |           0.055            |    3.176/2.137     |
> |    Uformer with GT-meanloss    |           **0.168**           |           **0.157**            |    **3.419/2.049**     |           **0.180**          |           **0.156**            |    **3.104/1.880**     |           **0.049**          |           **0.045**            |    **3.283/2.177**     |
> |           SNR-Aware            |           0.164           |           0.158            |    3.330/1.893     |           0.169           |           0.161            |    3.354/1.879     |           0.064           |           0.058            |    3.275/2.209     |
> |   SNR-Aware with GT-meanloss   |           **0.158**           |          **0.153**          |    **3.509/1.913**     |           **0.164**           |           **0.154**            |    **3.468/1.889**     |           **0.057**           |           **0.050**           |  **3.326**/~~2.207~~   |
> |            CID-Net             |           0.086           |           0.079            |    4.087/2.157     |                           |                            |                    |                           |                            |                    |
> |    CID-Net with GT-meanloss    |           **0.081**           |           **0.075**            |  ~~4.074~~/**2.161**   |                           |                            |                    |                           |                            |                    |

---

> ### Author Response · Authors · 2024-11-21
>
> GT-Mean Loss outperforms in  **all** average metrics than the baseline on unpaired datasets.
> |                                |      DICM       |                   |       MEF       |                   |      LIME       |                   |       NPE       |                   |       VV        |                   |     **AVG**     |                   |
> | :----------------------------: | :-------------: | :---------------: | :-------------: | :---------------: | :-------------: | :---------------: | :-------------: | :---------------: | :-------------: | :---------------: | :-------------: | :---------------: |
> |                                | MUSIQ $\uparrow$ | Q-Align $\uparrow$ | MUSIQ $\uparrow$ | Q-Align $\uparrow$ | MUSIQ $\uparrow$ | Q-Align $\uparrow$ | MUSIQ $\uparrow$ | Q-Align $\uparrow$ | MUSIQ $\uparrow$ | Q-Align $\uparrow$ | MUSIQ $\uparrow$ | Q-Align $\uparrow$ |
> |         RetinexFormer          |     57.398      |    3.800/2.740    |      56.17      |    3.111/2.323    |     57.262      |    3.111/2.323    |     60.507      |    3.673/2.699    |     37.513      |    3.471/2.154    |     53.770      |    3.438/2.458    |
> | RetinexFormer with GT-meanloss |     57.247      |    3.805/2.773    |     56.633      |    3.273/2.423    |     57.374      |    3.273/2.423    |     60.682      |    3.706/2.719    |     37.654      |    3.517/2.166    |   **53.918**    |  **3.490/2.498**  |
> |             MIRNet             |     52.467      |    3.111/2.337    |     47.399      |    2.860/2.088    |     54.837      |    2.860/2.088    |     58.641      |    3.285/2.374    |     54.566      |    2.955/2.162    |     53.582      |    2.991/2.203    |
> |    MIRNet with GT-meanloss     |     53.188      |    3.295/2.375    |     47.611      |    2.747/2.058    |     55.776      |    2.747/2.058    |     58.718      |    3.366/2.428    |     54.891      |    3.120/2.215    |   **54.037**    |  **3.069/2.225**  |
> |            LLFormer            |     56.642      |    3.379/2.526    |     53.335      |    2.836/2.102    |     55.671      |    2.836/2.102    |     59.824      |    3.445/2.551    |     60.885      |    3.067/1.955    |     57.271      |    3.079/2.225    |
> |   LLFormer with GT-meanloss    |     57.038      |    3.521/2.571    |     53.842      |    2.946/2.152    |      55.83      |    2.946/2.152    |     60.044      |    3.580/2.605    |     60.858      |    3.137/1.997    |   **57.522**    |  **3.178/2.268**  |
> |           Restormer            |     58.525      |    3.885/2.800    |     56.528      |    3.267/2.466    |     58.461      |    3.267/2.466    |     61.031      |    3.781/2.735    |     37.919      |    3.710/2.264    |     54.493      |    3.572/2.536    |
> |   Restormer with GT-meanloss   |     58.604      |    3.913/2.821    |     56.522      |    3.380/2.521    |     58.124      |    3.380/2.521    |     60.971      |    3.820/2.769    |      38.29      |    3.712/2.255    |   **54.502**    |  **3.607/2.559**  |
> |            Uformer             |     58.084      |    3.832/2.788    |     56.177      |    3.040/2.343    |     57.698      |    3.040/2.343    |      61.31      |    3.657/2.716    |     36.235      |    3.557/2.249    |     53.901      |    3.453/2.500    |
> |    Uformer with GT-meanloss    |     58.981      |    3.910/2.837    |     56.641      |    3.118/2.416    |      58.2       |    3.118/2.416    |     61.704      |    3.707/2.731    |     36.695      |    3.563/2.231    |   **54.444**    |  **3.505/2.528**  |
> |           SNR-Aware            |     47.025      |    2.971/2.144    |     48.685      |    2.646/1.967    |     49.216      |    2.646/1.967    |     46.441      |    2.938/2.131    |     23.186      |    2.904/1.839    |     42.911      |    2.798/2.005    |
> |   SNR-Aware with GT-meanloss   |      47.43      |    3.067/2.180    |      48.78      |    2.712/1.973    |     49.008      |    2.712/1.973    |     46.602      |    2.956/2.113    |     23.853      |    3.001/1.848    |   **43.135**    |  **2.861/2.012**  |

---

> ### Author Response · Authors · 2024-11-21
>
> - Q2:
> 	Our method actually enables the original baseline to better handle complex image degradation factors. Specifically, the first term in Eq. 1 ensures that the baseline retains its original performance, while the second term reduces the impact of brightness mismatch, allowing the baseline to better address these complex degradation factors. This is because, after GT-mean adjustment, the model learns to repair degradations at the same brightness level.
>
> - Q3:
> The computational cost of GT-Mean Loss is almost negligible:
>
> 	1. We provide a closed-form solution for $W$, enabling its fast computation.
> 	2. The second term in GT-Mean Loss is essentially the GT-mean-adjusted version of the original loss, with identical computational complexity. The original loss computation is minimal compared to the model’s forward and backward passes.
> 	To support this, we have provided an anonymized code repository: [https://anonymous.4open.science/r/Retinexformer-FEC9/](https://anonymous.4open.science/r/Retinexformer-FEC9/).
> 	For instance, training Retinexformer on LOLv 1 with a 4090 GPU takes 195 minutes for the baseline and only 199(**+4**)   minutes when using GT-Mean Loss.
>
> 	GT-Mean Loss is widely applicable to various existing methods without altering the baseline model structure. Despite its minimal cost, it delivers notable improvements: on LOLv 1/v 2, the average PSNR increases by +0.617, and SSIM improves by +0.013. These results are substantial given the simplicity and generalizability of our approach.
>
>
> - Q 4: Due to time constraints, we were unable to conduct a user study. However, we have provided Q-ALIGN metrics for all results to simulate human perception. The results indicate that our method is generally effective across all datasets.
>
> - Q 5:
> 	Thank you for the suggestion. We have revised the captions in **Appendix E** for the visualized results to include more detailed descriptions. We hope this improves clarity.
>
> 	**Specific changes:**
>
> 	*"The methods based on GT-Mean Loss achieve better exposure control in images 2-3, 8, and 10-13, enhancing the details in the dark areas to an appropriate level, while the baseline exhibits artifacts and color distortion due to overexposure in these images. In images 4 (the road in the lower left corner), 9 (the roof), and 14 (the palette), methods based on GT-Mean Loss provide more accurate colors. Additionally, methods based on GT-Mean Loss significantly suppress artifacts in images 1 and 5-8."*
> - Q 6 In fact, our motivation is clearly stated in the sixth paragraph of the Introduction:
> 	_"The issue of model training under brightness mismatch has largely been ignored in existing supervised LLIE research, despite some indirect solutions..."_
>
> *Looking forward to your feedback*

---

> ### Comment · Reviewer_5DCR · 2024-12-02
>
> The authors have addressed some of my concerns.

---

### Author Response · Authors · 2024-11-21

We sincerely thank all the reviewers for their insightful comments and valuable suggestions, which have been instrumental in guiding us to further improve our paper.

The primary contributions of this paper are as follows:

- Identification of the **brightness mismatch** issue, a phenomenon widely observed but often overlooked in the low-light image enhancement (LLIE) domain, and an explanation of how this issue impacts training and testing of low-light images.
- Based on this discovery, we propose **GTmean loss**, a simple yet effective solution to address this problem:
    - It is **conceptually straightforward**, with W computed using KL divergence.
    - It is **widely applicable**, imposing minimal constraints on the original loss function used to construct GTmean loss, requiring only input images and their corresponding ground truths (GTs).
    - It has **low implementation cost**, with training times almost identical to the original baseline in practice.
    - It is **highly effective**, consistently improving baselines as demonstrated across extensive experiments.

Finally, we advocate for the LLIE community to report **both normal metrics and GTmean metrics simultaneously**, rather than reporting GTmean metrics in isolation. This will enable a more comprehensive evaluation of low-light image performance.

---

We have made every effort to address all concerns by providing detailed clarifications and requested results. Below is a summary of the major revisions:

**Main text:**

- Refined the caption for Fig. 2.
- Refined the description of GTmean loss in Section 3.1.
- Corrected typos.
    (_Thanks to reviewers 25iy and rJj2, these revisions have improved the clarity and readability of our paper._)

**Appendix:**

- Added a deeper analysis of W in Appendix A.
- Included additional quantitative results in Appendix D, covering metrics such as LPIPS, MUSIQ, and Q-Align, all of them demonstrate the superiority of our method over the baseline.
- Revised captions for visualizations in Appendix E.
    (_Thanks to reviewers 5DCR and LK8R, the new analyses and experimental results enhance the soundness of our work._)

**Code:**
We have integrated GTmean loss into the BasicSR framework and provided the code and checkpoints for public access.
(_Thanks to reviewer wA78 for suggesting open-sourcing our work._)
[https://anonymous.4open.science/r/Retinexformer-FEC9/](https://anonymous.4open.science/r/Retinexformer-FEC9/)

---

### Author Response · Authors · 2024-11-30

Dear Reviewers,

Thank you again for your valuable feedback. We have carefully addressed your comments in the rebuttal and revised the manuscript accordingly.

We kindly invite you to review the updates, and please let us know if you have any further questions or suggestions. Your time and insights are greatly appreciated.

Best regards,

Authors

---

### Note · Authors · 2025-06-26

I have read and agree with the venue's withdrawal policy on behalf of myself and my co-authors.

---

### Meta-Review · Area_Chair_aJgB · 2024-12-16

**Metareview:**

The paper was reviewed by five experts. They are unanimously recommend rejection.

The authors provided responses, addressed part of the raised concerns, but this was found insufficient and none of the reviewers improved their ratings to the acceptance range.

The AC agrees with the reviewers that the paper is marginally below the acceptance threshold and invites the authors to benefit from the received feedback and to further improve their work.

**Additional Comments On Reviewer Discussion:**

All the reviewers provided negative initial ratings and the provided responses where found insufficient by them to improve their ratings while agreeing that part/some of the initial concerns were addressed.

---

### Decision · Program_Chairs · 2025-01-22

Reject